# Peculiar transient behaviors of organic electrochemical transistors governed by ion injection directionality

Ji Hwan Kim [1,6], Roman Halaksa[2,6], Il-Young Jo[1], Hyungju Ahn[3], Peter A. Gilhooly-Finn[2], Inho Lee[4], Sungjun Park [4,5], Christian B. Nielsen [2] ✉ & Myung-Han Yoon[1] ✉

Despite the growing interest in dynamic behaviors at the frequency domain, there exist very few studies on molecular orientation-dependent transient responses of organic mixed ionic–electronic conductors. In this research, we investigated the effect of ion injection directionality on transient electrochemical transistor behaviors by developing a model mixed conductor system. Two polymers with similar electrical, ionic, and electrochemical characteristics but distinct backbone planarities and molecular orientations were successfully synthesized by varying the co-monomer unit (2,2'-bithiophene or phenylene) in conjunction with a novel 1,4-dithienylphenylene-based monomer. The comprehensive electrochemical analysis suggests that the molecular orientation affects the length of the ion-drift pathway, which is directly correlated with ion mobility, resulting in peculiar OECT transient responses. These results provide the general insight into molecular orientation-dependent ion movement characteristics as well as high-performance device design principles with fine-tuned transient responses.

Organic mixed ionic–electronic conductors (OMIECs) are a promising class of materials suitable for non-conventional applications such as bioelectronics[1–4], neuromorphic transistors[5–7], and bio-fuel cells[8], where both ionic and electronic conduction play critical roles. For these purposes, application-specific characteristics should be incorporated into OMIECs in combination with prominent ionic and electronic conductivities, for instance, high sensitivity/selectivity of target analytes, functional groups for tethering enzymes/mediators, and appropriate responses to high frequency input signals. Accordingly, the development of methods for evaluating such characteristics is as important as designing/synthesizing advanced OMIECs with improved figure-of-merits (i.e., $\mu C^*$). One of the very effective device platforms for characterizing the comprehensive conduction of ions and electrons/holes is organic electrochemical transistor (OECT)[9–11]. The channel conductivity of an OECT device is modulated by the gate-bias-induced ion migration through the active layer, which is in direct contact with electrolytes. During the channel conductivity modulation, various parameters such as charge carrier mobility, ion mobility, ion–polymer backbone coupling efficiency, and solvent-induced swelling in the channel region determine the ion/electron transduction efficiency.

Various types of OMIECs have been intensively investigated using the OECT platform or the OECT device combined with other probing techniques, and it has been suggested that the direction[12], length[13], symmetry of side chain[14], the molecular structure of polymer backbone[15], and film crystallinity-related factors affect the

[1]School of Materials Science and Engineering, Gwangju Institute of Science and Technology (GIST), Gwangju 61005, Republic of Korea. [2]Department of Chemistry, Queen Mary University of London, London E1 4NS, UK. [3]Pohang Accelerator Laboratory, Pohang 37673, Republic of Korea. [4]Department of Intelligence Semiconductor Engineering, Ajou University, Suwon 16499, Republic of Korea. [5]Department of Electrical and Computer Engineering, Ajou University, Suwon 16499, Republic of Korea. [6]These authors contributed equally: Ji Hwan Kim, Roman Halaksa. ✉e-mail: c.b.nielsen@qmul.ac.uk; mhyoon@gist.ac.kr

corresponding OECT characteristics[16–18]. These studies not only showed new phenomena such as electrolyte-mediated swelling in the OECT channel but also the verification of existing principles originally derived from organic field-effect transistors, organic photovoltaics, and organic light-emitting diodes. Recently, several in situ OECT studies dealing with volumetric expansion by water molecules and ionic species have demonstrated the effect of side chain and backbone structures of OMIECs on OECT behaviors by systematically introducing molecular design variations[19–24]. It has been reported that the crystallographic features induced by molecular structural factors significantly affect various OECT characteristics[25–28]. Nonetheless, the majority of these studies have mainly focused on the relation between structural properties and *steady-state* device characteristics (e.g., charge carrier mobility, volumetric capacitance, $\mu C^*$), while there exist only a few reports on their influence on transient characteristics which are critical for rapid responses and/or frequency-dependent features in special applications such as neural interfaces, neuromorphic transistors, drug delivery devices. Recently, Rivnay and coworkers have reported the time-resolved structural change in highly crystallized poly(3,4-ethylenedioxythiophene):poly(sytrenesulfonate) (PEDOT:PSS) films under electrical bias[29]; however, they mainly focused on the fundamental aspects of this phenomenon rather than the general method for evaluating transient characteristics of a given OMIEC.

In this research, two 1,4-dithienylphenylene(DTP)-based polymers with four triethylene-glycol side chains per repeating unit were successfully synthesized to investigate the correlation between backbone planarity-dependent molecular orientation and transient OECT characteristics. The microstructural, optical, and electrochemical properties of each DTP polymer film were examined by grazing incidence wide angle X-ray diffraction (GIWAXD), ultraviolet-visible (UV-vis) spectroscopy, cyclic voltammetry (CV), and spectroelectrochemical analysis to investigate the effect of the molecular structure and orientation of the polymer backbone on optical and electrochemical properties. An orthogonal patterning process using cyclic transparent optical polymer (CYTOP™) was employed to protect the active channel layer during the OECT fabrication, and the corresponding device characteristics such as transconductance, $\mu C^*$ and hole mobility were extracted. Simultaneously, volumetric capacitances at different offset voltages were measured through electrochemical impedance spectroscopy (EIS), revealing the correlation between the chemical/electrochemical properties of the DTP-based polymers and the corresponding OECT characteristics. Furthermore, the molecular orientation-dependent ion mobilities and transient responses of these polymers were investigated by moving front and transient frequency-dependent measurements, while these results were interpreted along with the microstructural analysis based on X-ray diffraction. In parallel, a different patterning method was introduced for controlled ion injection into the active layer of OECT in order to verify that the ion injection direction relative to the molecular orientation in the active channel layer is the key factor in determining transient behaviors of OMIECs.

## Results

### Structural characterizations

Figure 1a shows a schematic representation of ionic−electronic mixed conducting polymers. In general, the conjugated hydrophobic polymer backbone facilitates the transport of electrons or holes, while the hydrophilic glycol side chains attached to the polymer backbone facilitate ion/water molecule penetration into the polymeric film. This allows for an alternating structure of hydrophilic and hydrophobic regions in a polymeric film, and makes the orientation of the polymer backbone a critical factor for ion migration in OMIECs. Figure 1b shows the molecular structures of the DTP-based semiconducting polymers with phenylene (DTP-P) and 2,2′-bithiophene (DTP-2T) as co-monomer units. The DTP unit was developed as a highly soluble and hydrophilic moiety due to its four triethyleneglycol side chains while also embedding a high degree of structural planarity around the conjugated motif due to favorable S-O interactions between the flanking thiophenes and the central doubly glycolated phenylene unit. As discussed further in detail in the Supporting Information, the DTP-monomer with four triethylene glycol side chains was synthesized from readily available starting materials in three steps with an overall yield of 32%. Subsequent Pd-catalyzed direct arylation reactions with 1,4-dibromobenzene and 5,5′-dibromo-2,2′-bithiophene provided DTP-P and DTP-2T, respectively (Fig. 1b); the direct arylation protocol was chosen over Stille polymerization due to its reduced number of synthetic steps, better atom efficiency and more benign reaction conditions[30]. The crude polymers were purified by Soxhlet extractions with hexane, methanol, and acetone. Thereafter, chloroform was used to isolate the purified materials with weight-average molecular weights

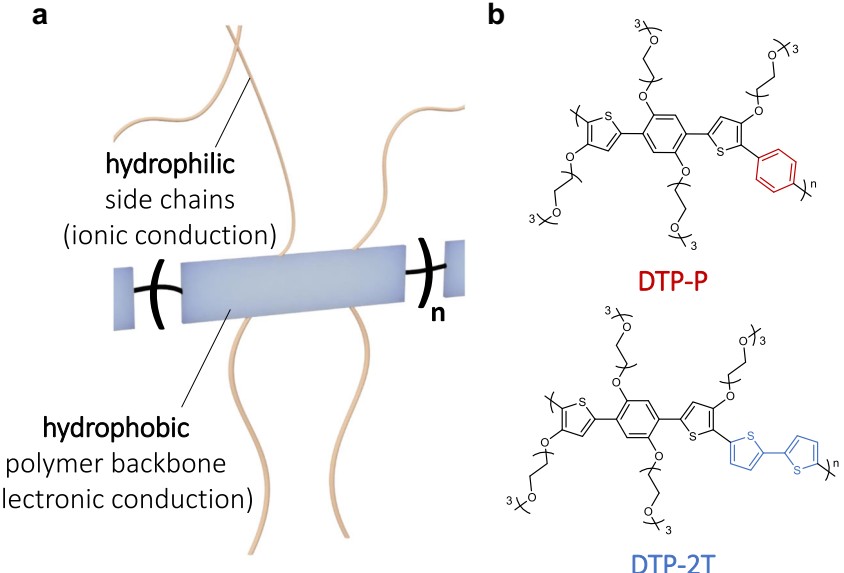

**Fig. 1 | Molecular structural characteristics of DTP-based mixed ionic−electronic conducting polymers. a** Schematic illustration of a general molecular structure of mixed ionic−electronic conducting polymers with glycol side chains. **b** Chemical structures of DTP-based mixed ionic−electronic conducting polymers (DTP-P and DTP-2T).

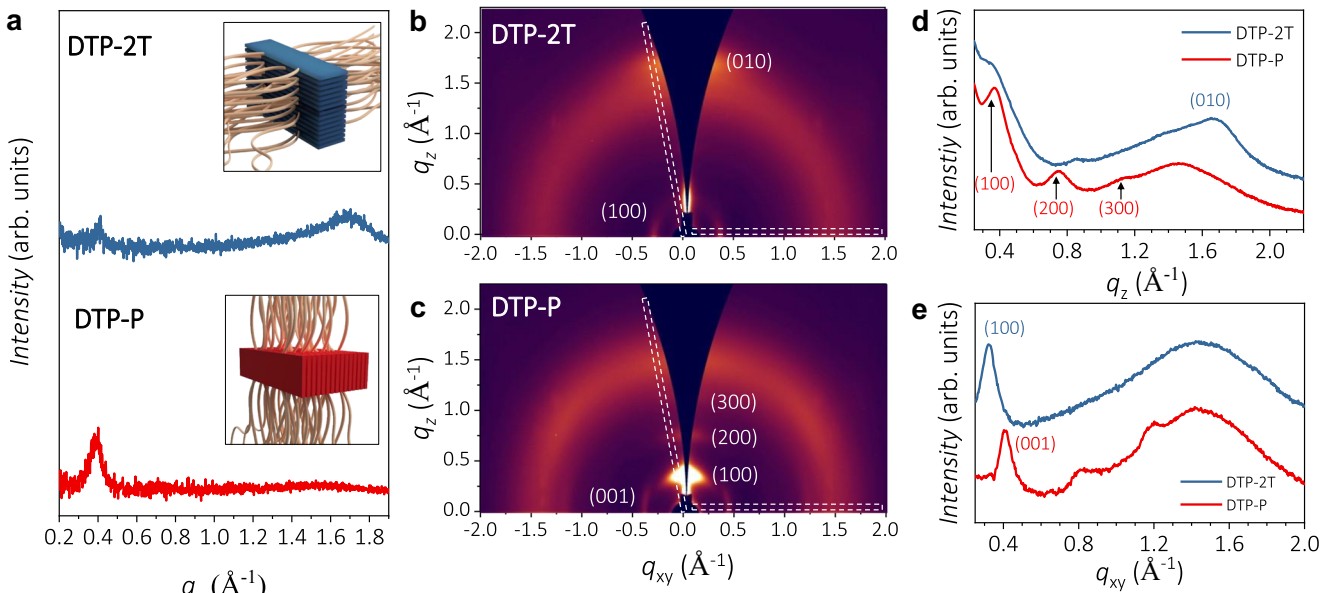

**Fig. 2 | Crystallographic microstructural analysis of DTP-based mixed ionic −electronic conducting polymers (DTP-P and DTP-2T). a** XRD patterns of DTP-P and DTP-2T films. The insets illustrate the dominant molecular orientations of DTP-P and DTP-2T, that is, *edge-on*, and *face-on*, respectively, relative to a substrate. 2D GIWAXD pattern images of spin-coated **b** DTP-2T and **c** DTP-P. GIWAXD **d** out-of-plane ($q_z$) and **e** in-plane ($q_{xy}$) line-cut profiles of DTP-2T and DTP-P films. The 1D line-cut profiles were obtained from the 2D GIWAXD regions defined by dashed white boxes.

of 42 kg mol⁻¹ (dispersity 1.9) and 45 kg mol⁻¹ (dispersity 2.8) for DTP-P and DTP-2T, respectively[31].

Based on the structural analysis below, the edge-on orientation is dominant in the DTP-P film, while the face-on orientation is dominant in the DTP-2T film (Fig. 2a). The choice of the co-monomer unit and the resulting difference in the planarity of each polymer backbone results in the orientational difference of the polymer backbone, which is supported by dihedral angle analysis using density functional theory (DFT) calculations (Supplementary Fig. 1). The phenylene co-monomer unit causes a significant backbone twist owing to steric crowding from the phenyl hydrogens interfering with the adjacent glycolated thiophene units. On the other hand, the bithiophene co-monomer adapts a significantly more coplanar conformation due to the smaller ring size, fewer space-filling hydrogens, and favorable S–O interactions which promote the coplanar s-trans conformation across all thiophene-thiophene couplings along the polymer backbone. These factors tend to govern the polymer solubility and polymer chain interactions during crystallite growth upon solution deposition and eventually crystallite orientation relative to the underlying substrate, that is, less coplanar DTP-P with an edge-on orientation and more coplanar DTP-2T with a face-on orientation, which is in agreement with similar observations for related conjugated systems[32–34]. In addition, this orientational difference became significantly more pronounced in the GIWAXD patterns of spin-coated DTP films (Fig. 2b, c). The π–π stacking peaks at higher *q* value in the out-of-plane are observed for DTP-2T, while lamellar stacking peaks were observed out-of-plane for DTP-P, suggesting the dominant molecular orientation of DTP-2T and DTP-P is face-on and edge-on, respectively (Fig. 2d). Interestingly, at lower *q* value in the in-plane, diffraction peaks were found corresponding to spacings of 19.5 and 15.3 Å for DTP-2T and DTP-P, respectively (Fig. 2e). It is straightforward to assign the in-plane peak observed from the DTP-2T as (100), while the in-plane peak of DTP-P does not match the lamellar peak distance obtained from the out-of-plane measurements. This in-plane peak is instead attributed to the polymer repeat unit length (001), based on the previous literature[35]. Therefore, it is not relevant to the dominant molecular orientation of the polymer film. In the 2D GIWAXD pattern of both DTP-2T and DTP-P, a broad peak corresponding to a spacing of ~4.2 Å was observed,

indicating the presence of SiO₂ substrate[36]. We also compared the pole figures of two polymers (Supplementary Fig. 2) and concluded that DTP-P exhibit mixed orientations, but the edge-on feature is rather dominant over the face-on feature.

## Electrochemical characterizations

As shown in the UV-vis absorption spectra (Fig. 3a) and Table 1, the absorption maxima of both the DTP polymers revealed a red shift of ~30 nm and the appearance of vibronic splitting from 0−0 and 0−1 optical transitions after film formation. These suggest the formation of an aggregated polymer backbone after spin-casting of the polymer solution. In addition, DTP-2T showed a higher absorption onset owing to the additional thiophene moieties in the polymer repeating unit. The red shift in the case of DTP-2T and the narrower optical band gap of DTP-2T compared to that of DTP-P are attributed to the stronger quinoidal contribution from thiophene, resulting in a higher degree of planarity and thus a longer effective conjugation length[37]. The cyclic voltammograms (CV) obtained from each polymer film in a 0.1 M NaCl aqueous solution are shown in Fig. 3b. Both DTP polymers can be oxidized and reduced in aqueous electrolytes, which is evident from the anodic and cathodic waves represented in CV (Fig. 3b). Ionization potentials (IPs) were estimated from the onset of oxidation in the cyclic voltammograms of polymer films with 0.1 M TBAPF₆ in acetonitrile as supporting electrolytes. DTP-2T has an IP of ~4.20 eV, whereas DTP-P has a higher IP of ~4.44 eV (Fig. 3c and Supplementary Fig. 3). The DFT calculations of the highest occupied molecular orbital (HOMO) energy levels were in good agreement with the data obtained from the CV (Supplementary Fig. 4). It is noteworthy that upon doping, the current density of DTP-2T becomes significantly higher than that of DTP-P. This is possibly attributed to the fact that the proportion of electroactive polymer backbone (relative to the side chains) is higher in DTP-2T than in DTP-P which can be rationalized by considering one polymer repeat unit with four triethylene glycol side chain across five aromatic rings in DTP-2T (averaging 0.8 side chain per aromatic unit) and four aromatic units in DTP-P (1 side chain per aromatic unit). Furthermore, the electron-rich thiophene group is beneficial for charge storage when ions compensate for the charges of the polymer backbone.

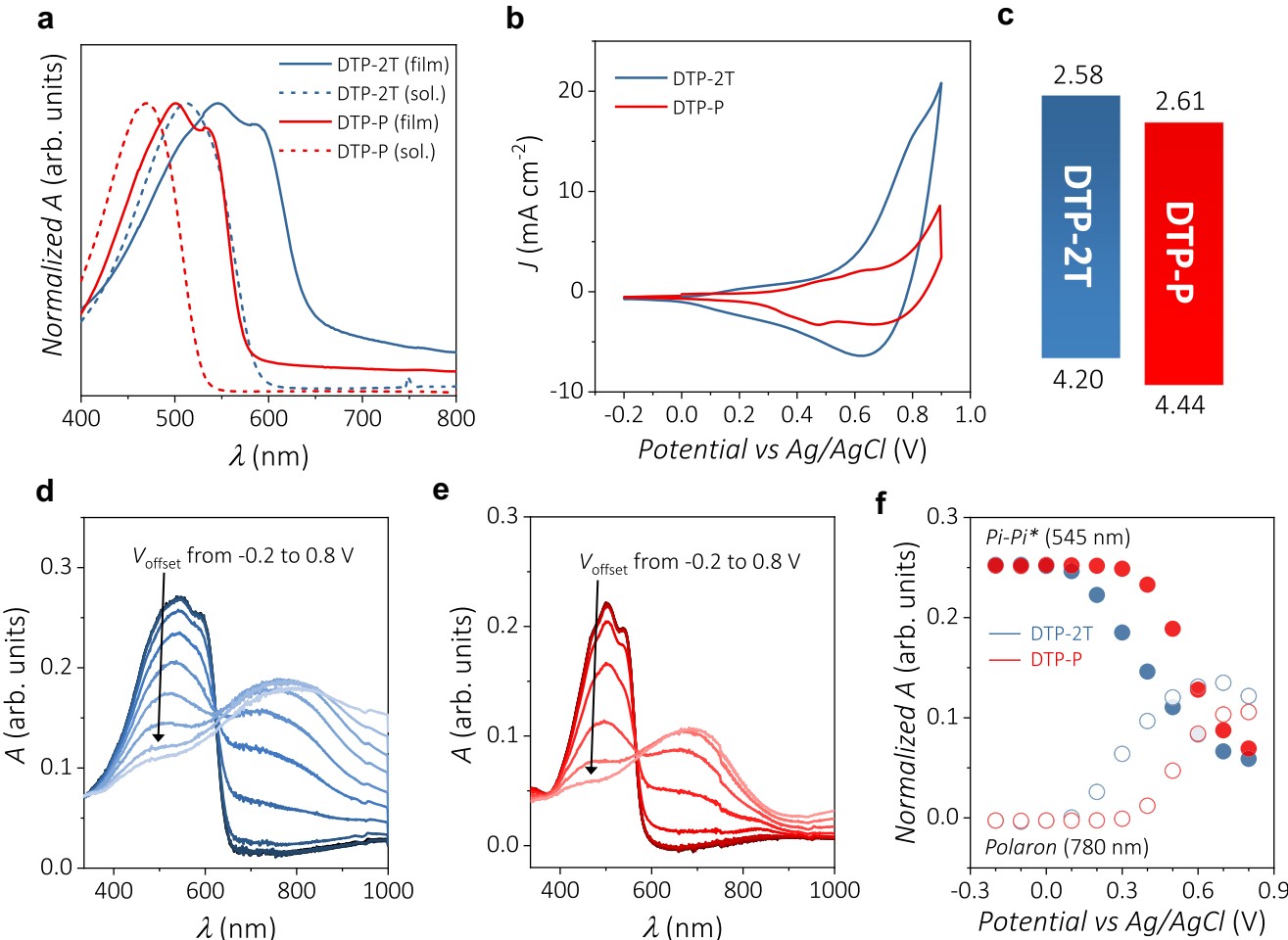

**Fig. 3 | Electrochemical characterizations of DTP-2T and DTP-P films. a** UV-vis absorption spectra of DTP-2T and DTP-P in a chloroform solution and as thin films prepared by spin casting. **b** Cyclic voltammograms of the DTP-2T and DTP-P thin films recorded in a 0.1 M NaCl$_{(aq)}$ electrolyte with a scan rate of 50 mV s$^{-1}$. **c** Ionization potentials (bottom) and electron affinities (top, estimated from ionization potential and optical band gap) in eV for DTP-2T and DTP-P. Spectro-electrochemical analysis results of **d** DTP-2T and **e** DTP-P using a 0.1 M NaCl$_{(aq)}$ electrolyte. **f** Normalized absorbance of π–π* transition and polaron peak obtained from DTP-2T and DTP-P polymer films with different offset voltages.

Spectroelectrochemistry was used to compare the oxidation characteristics of the polymers. Figure 3d, e show the spectro-electrochemical analysis of the DTP-2T and DTP-P thin films, respectively. Both polymers showed an electrochromic response during the potential sweep from −0.2 to 0.8 V versus Ag/AgCl. For DTP-2T, from 0.1 V, the gradual suppression of the π–π* transition peak accompanied by the appearance of an absorption peak around 780 nm is attributed to polaron formation. Further increasing the potential to 0.7 V decreased the polaron absorption intensity, which could be explained by the formation of bipolaron represented by longer wavelength absorption[38]. A bias of −0.2 V was re-applied to confirm the reversibility of the doping process. The absorption intensity of the π–π* transition was fully restored, indicating that there was no degradation during the electrochemical cycling of the polymer.

Identical measurements were conducted on DTP-P, and the whole doping process was reproduced except for the oxidation starting potential, which was 0.3 V (Fig. 3f) in the case of DTP-P, due to the difference in the HOMO level of each polymer. This reproducible stable electrochromic behavior suggests that these polymers are suitable for OECTs to modulate the electronic current using doping/dedoping cycling induced by ionic current.

## Steady-state OECT performance

Microchannel OECTs of dimensions 80 μm × 20 μm were fabricated using a orthogonal patterning process without the parylene patterning process, which is a common method for OECT fabrication[14,19,39,40]. The detailed procedure for OECT fabrication can be found in the Supplementary Fig. 5. All devices were characterized without further annealing, while 0.1 M NaCl aqueous electrolyte and a Ag/AgCl electrode as the gate electrode were employed. All parameters relevant to the OECT devices, such as OECT active layer thickness ($d$), transconductance ($g_m$), volumetric capacitance ($C^*$), OECT mobility ($\mu_{OECT}$), threshold voltage ($V_{Th}$), $\mu C^*$ product, and on/off current ratio are summarized in Table 2. Detailed measurement procedures are also provided in the Supporting Information (Supplementary Figs. 6 and 7). The $\mu C^*$ values estimated from the orthogonally patterned devices were comparable to those of the parylene-patterned devices while their variations are within the standard deviation (Supplementary

## Table 1 | Optoelectronic properties of DTP-2T and DTP-P

| Polymer | $E_{ox,aq}$ (V)[a] | IP (eV) | $\lambda_{max,soln}$ (nm)[b] | $\lambda_{max,film}$ (nm)[c] | $E_{g,opt}$ (eV)[d] |
|---|---|---|---|---|---|
| DTP-2T | 0.06 | 4.20 | 512 | 545 | 1.95 |
| DTP-P | 0.36 | 4.44 | 470 | 501 | 2.18 |

[a]0.1 M solution of NaCl in H$_2$O used as electrolyte.
[b]Measured in chloroform.
[c]Recorded while applying 0 V.
[d]Calculated from the onset of absorption.

**Table 2 | Steady-state OECT performance**

| Polymer | $d$ (nm) | $g_m$ (µS) | $C^{*a}$ (F cm$^{-3}$) | $\mu_{OECT}$[b] (cm$^2$ V$^{-1}$ s$^{-1}$) | $V_{Th}$ (V) | $\mu C^{*c}$ (F V$^{-1}$ cm$^{-1}$ s$^{-1}$) | Current on/off |
|---|---|---|---|---|---|---|---|
| DTP-2T | 101 ± 2 | 41 ± 2 | 166 ± 15 | 0.39 | − 0.50 ± 0.01 | 65 ± 11 | ~10$^3$ |
| DTP-P | 165 ± 2 | 19 ± 4 | 113 ± 12 | 0.61 | − 0.63 ± 0.01 | 71 ± 10 | ~10$^3$ |

[a]Values calculated from the Bode plot obtained from EIS measurements.
[b]Calculated by dividing the $\mu C^*$ values with the respective $C^*$ values.
[c]Average value obtained from five different channels from the slope of $g_m$ as a function of $(Wd/L)(V_{Th} − V_G)$.

Fig. 6). It is noteworthy that the newly synthesized OMIECs can be incorporated into the OECTs fabricated using the solution-based CYTOP patterning method even when the vacuum-based parylene coater is not available for patterning the OMIEC channel layer which is typically not compatible with the conventional photolithography process.

The representative output curves and saturation-regime transfer curves of all OECT devices were plotted as shown in Supplementary Fig. 8. All devices showed typical p-type accumulation mode operation, indicating that the channel conductivity, which is generally low due to the semiconducting nature of the polymers, was increased as ions were injected into the channel; thus, the conjugated cores of the polymers were doped[9]. With a very low gate leakage current level, the transfer curves indicate comparable on-current values in the doped state for two polymers. The threshold voltage ($V_{Th}$) difference between the two polymers was -0.13 V, which also reflects the order of the measured oxidation potentials of the polymers. For each polymer, the transconductance ($g_m$), that is, the rate of change in $I_D$ with $V_G$, is plotted in Supplementary Fig. 8c. It can be seen that the highest $g_m$ values were obtained at a gate voltage ($V_G$) = −0.8 V. Aqueous electrolyte limits the voltage window to less than −0.8 V due to the electrolysis of water; hence, the $V_G$ sweep range cannot be exceeded over the $V_G$ range showing peak $g_m$ value. Nonetheless, we took the highest $g_m$ value to obtain the figure of merit for comparing OMIECs after confirming that the $g_m$ values of both polymers at $V_G$ −0.8 V were sufficiently saturated (Supplementary Fig. 9). For fair comparison, the $g_m$ values should be normalized by a geometrical factor and biasing condition because $g_m$ is obtained from the product of both these factors, as shown in Eq. 1.

$$g_m = \frac{\partial I_D}{\partial V_G} = \mu C^* \frac{Wd}{L}(V_G - V_{Th}) \tag{1}$$

where $\partial I_D$ is the differential source-drain current, $\partial V_G$ is the corresponding source-gate voltage differential, $\mu$ is the electronic charge carrier mobility, $C^*$ is the volumetric capacitance, $W$ is the channel width, $d$ is the channel thickness, $L$ is the channel length, $V_{Th}$ is the threshold voltage, and $V_G$ is the gate voltage. By fabricating OECTs with channel lengths varying from 20 to 80 µm and plotting the corresponding $g_m$ values as a function of $(Wd/L)(V_G - V_{Th})$ (Figs. S8d and S10), it was found that DTP-2T and DTP-P have very similar figure-of-merit ($\mu C^*$) values.

To decouple $C^*$ and hole mobility ($\mu_h$), EIS measurements were conducted for both the polymers. By fitting the impedance data to an R(RC) circuit, the capacitance value of each polymer was obtained from a Bode plot at different offset voltages ($V_{offset}$). Thereafter, $C^*$ was obtained by dividing the capacitance value by the nominal volume of the polymer film for EIS measurement. Supplementary Fig. 8e shows that $C^*$ increases and saturates as $V_{offset}$ changes from −0.4 to 0.8 V, representing the doping of the semiconducting polymers. The highest $C^*$ values of DTP-2T and DTP-P were 166 and 113 F cm$^{-3}$, respectively. Finally, the OECT mobility was decoupled from the $\mu C^*$ value, and DTP-P showed higher hole mobility (0.61 cm$^2$ V$^{-1}$ s$^{-1}$) than DTP-2T (0.39 cm$^2$ V$^{-1}$ s$^{-1}$) (Supplementary Fig. 8f). Considering electrochemical doping/dedoping minimally affected the dominant molecular

orientations in these polymer films[31], this trend could be attributed to the edge-on dominant mixed orientations in DTP-P, which is still favorable for lateral charge transport compared with the predominant face-on orientation[37,41,42]. Although the $\mu C^*$ values of these two polymers are similar, the molecular orientations of these two polymers are very different (i.e., edge-on dominant DTP-P vs. face-on dominant DTP-2T), which suggests that these two polymers are the best model system to investigate the effect of molecular orientation on the transient response of OECTs while the overall figure-of-merits remain similar.

### Ion mobility and transient OECT characteristics

To investigate the effect of molecular orientation on the dynamic response of OMIECs, we conducted moving front experiments which are well known for determining the one-dimensional ion mobility in mixed conducting polymer films. As shown in Fig. 4a, the mixed conducting polymer layers were deposited on a glass substrate and passivated by photoresist to prevent vertical ion injection. Moreover, the NaCl electrolyte and polymer film formed a planar junction for the lateral injection of ions. A short movie clip was recorded while applying the voltage, and several time spot images showing time-dependent color change were obtained. The corresponding movie clips from the moving front experiments of DTP-P and DTP-2T can be found in Supplementary Movies 1 and 2, respectively. The intensity profile of each image was determined using ImageJ software and plotted as shown in Fig. 4b. In the intensity profile plot, the color change front is represented as a drastic intensity change; therefore, the time-dependent position of the color change front can be obtained by extrapolating the slope of the drastic intensity change and calculating the x-intercept of the extrapolated line. The distance from the initial position of the color change was plotted as a function of time (Fig. 4c). The ionic velocities of DTP-2T and DTP-P were 17.4 and 1.8 µm s$^{-1}$, respectively. Finally, the ion mobility was obtained using the following equation that describes the relationship between the applied electric field and the velocity of the ion:

$$v_{ion} = \mu_{ion} E \tag{2}$$

where $v_{ion}$ is the velocity of the ion, $\mu_{ion}$ is the ion mobility, and $E$ is the electric field applied throughout the channel of the "moving front" experimental setup. The measured ion mobility of DTP-2T was approximately 10 times higher than that of DTP-P (Fig. 4d). This clearly indicates that the face-on orientation is highly advantageous for ion drift in lateral direction (ion drift in parallel with the lamellar stacking direction [100]) throughout the mixed conducting polymer film compared with the edge-on orientation (ion drift in parallel with the π–π stacking direction [010]).

Another way to demonstrate the dynamic response of the OMIECs is transient behavior measurement under a constant gate voltage. As shown in Fig. 4e, we used a function generator to apply a gate voltage pulse, and then the transient response of the drain current was measured using a source meter, current preamplifier, and data acquisition system. In the transient response measurement, the geometry of active layer is important factor affecting transient response (Fig. 4f, left panel). As shown in the middle panel of Fig. 4f, the ions can be injected through the whole surface of active layer except the bottom surface,

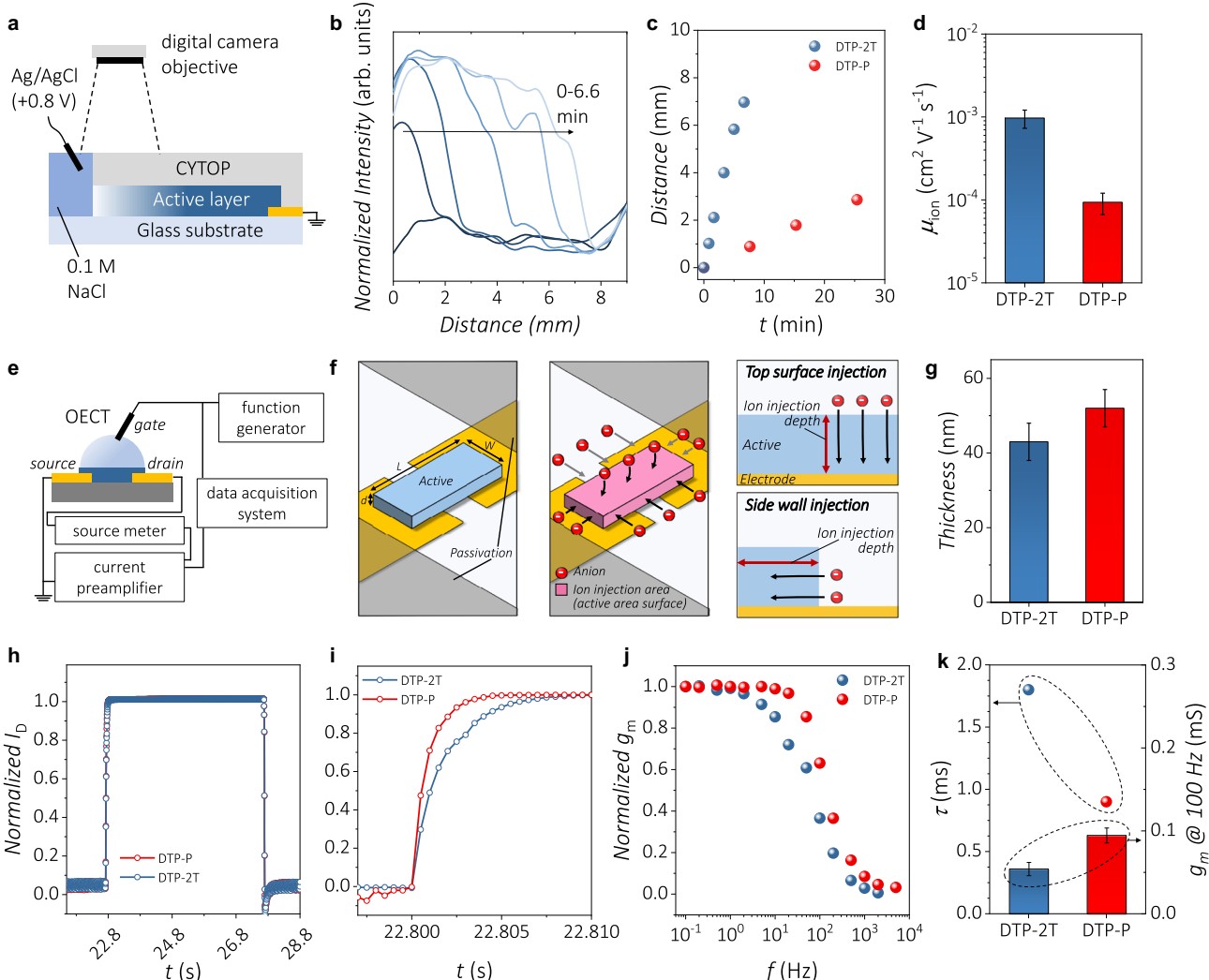

**Fig. 4 | Ionic mobility and transient behaviors of DTP-2T and DTP-P films.**
**a** Schematic illustration of the moving front experiment to obtain ion mobility.
While applying bias, the propagation of the dedoping front was recorded through
time-lapse imaging of the DTP-2T and DTP-P films. **b** One-dimensional normalized-
intensity profiles as a function of distance from the electrolyte–polymer interface
(from $t$ = 0 to 6.6 min) are obtained from the time-lapse images. **c** Distance from the
electrolyte–polymer interface obtained from each intensity profile at certain point
of time. The slope of each graph represents the drift velocity of ions in DTP-2T and
DTP-P. **d** Plot of ion mobility in the DTP-2T and DTP-P films. The measurements
were performed using more than five devices as each set. **e** Schematic illustration of
transient and frequency responses setup. **f** Schematic definition of the geometrical
parameters in the OECT active layer. $W$, $d$, and $L$ denote the width, thickness and
length of the active layer, respectively. Middle panel shows the ion injection area of
active layer. Pink colored area represents the ion injection area, which is active layer

surface. Right panel shows the ion injection depth of active layer. Pink arrow
represents the depth of ion injection depending on the ion injection direction. **g**
Thickness of active layer in OECT for transient measurement. Thickness values
were obtained by measuring more than five devices for each set. **h** Transient
response of the drain current at a constant $V_D$ of −0.6 V and a square voltage pulse
of 5 s ($V_G$ = −0.8 V) at the gate electrode. **i** Initial time range of transient response of
the drain current. **j** Plot of frequency-dependent normalized transconductance
obtained from frequency response measurements at $V_D$ = −0.6 V. A sine gate vol-
tage pulse is applied at the gate electrode using a function generator. **k** Plot of rise
time constant obtained from transient response (left $y$-axis) and the plot of trans-
conductance obtained by the application of $V_G$ sine wave at a frequency of 100 Hz
(right $y$-axis). Transconductance values were obtained from more than five devices
for each set.

which is substrate–active layer interface. The active layer geometry
also affects the ion injection depth depending on the ion injection
direction (right panel of Fig. 4f). Therefore, the OECT devices for
transient and frequency response studies were fabricated with com-
parable active layer thickness (DTP-2T: 43 ± 5 nm, DTP-P: 52 ± 5 nm) in
order to minimize the effect of active layer thickness on resultant
device characteristics for fair comparison (Fig. 4g). The results showed
that there was a small difference in the transient response of the drain
current (Fig. 4h, i, Supplementary Fig. 11, and Supplementary Table 1)
for the DTP-P and DTP-2T active layers. For quantitative comparison of
this transient response, the rise time constant was obtained by fitting
with an exponential decay function, as described by the equation

below:

$$I_D(t) = I_{D,0} + A \times \exp(-t/\tau) \tag{3}$$

where $I_D(t)$ represents the $I_D$ at time $t$ after applying the gate bias, $I_{D,0}$
represents the initial $I_D$ before applying the gate bias, $A$ is a constant,
and $\tau$ is the time constant. The rise time constants for DTP-2T and DTP-
P were estimated to be 1.8 and 0.9 ms, respectively. These results
demonstrate that the transient response measurement is effective for
examining the ion drift behavior considering that the time scale of ion
drift is too fast to be observed by conventional (steady-state) cyclic
voltammetry and spectroelectrochemistry.

For the more cautious comparison, the geometry-normalized time constants[43] were calculated by dividing $\tau$ with $d(WL)^{1/2}$ to eliminate the possible effect of geometrical factor difference such as active layer thickness and ion injection area by considering that $\tau$ is proportional to $d(WL)^{1/2}$. The normalized time constant values for DTP-2T and DTP-P were calculated as $9.8 \pm 1.7 \times 10^8$ and $5.3 \pm 0.3 \times 10^8$ s m$^{-2}$, respectively, showing opposite trend observed from ionic mobility results. This can be explained by the ion injection direction of each device. Considering the OECT devices (Fig. 4f), the ions are injected into the mixed conducting polymer film from the whole surface of the film, except for the bottom surface. However, the ions in the "moving front" experiment (Fig. 4a) are injected from the side wall of the polymer film (lower right panel of Fig. 4f). Therefore, the vertical ion injection is dominant for these OECTs, while the lateral ion injection is dominant in the case of "moving front" devices due to the difference in ion injection direction. This reveals that the ion injection is significantly faster when the ion injection direction is in parallel with the lamellar stacking direction [100]. The frequency response results also support the concept of ion-injection-direction-dependency. As demonstrated in Fig. 4j, the $g_m$ of the OECTs was maintained until the cut-off frequency was reached, at which $g_m$ starts to decay abruptly. To quantitatively compare the frequency responses of DTP-2T and DTP-P, the $g_m$ values of each polymer at a frequency of 100 Hz were compared (Fig. 4k, column graph). The $g_m$ values of DTP-2T and DTP-P were reduced to 36% and 63%, respectively, of the corresponding $g_m$ values calculated at a frequency of 0.1 Hz, verifying that the dynamic behavior of DTP-P is significantly faster than that of DTP-2T in this device configuration. Because the OECT device configuration in which vertical ion injection is dominant was used for the frequency response measurement, the edge-on dominant mixed conducting polymer DTP-P showed a faster ionic response than the face-on oriented DTP-2T polymer.

## Molecular orientation-dependent transient behaviors

Although DTP-2T and DTP-P showed that ionic mobilities and frequency-dependent transient behaviors of OECTs depend on the direction of ion injection, there exists still the possibility of material-dependent characteristics (e.g., molecular weight, relative crystallinity) regardless of the molecular orientation of OMIECs. Finally, different patterning methods were employed to demonstrate the molecular orientation dependency of the transient behavior. As shown in Fig. 5a, the parylene patterning and orthogonal (CYTOP) patterning methods were employed for both DTP-2T and DTP-P. The schematic representation of rather simplified molecular orientations in DTP-2T and DTP-P films is provided as the polymers take the designated preferential orientation for the better understanding of apparent charge/ion transport in terms of the relation between molecular orientation and ion injection direction. The detailed device fabrication procedure of these devices is described in Supplementary Fig. 12. In the case of the parylene patterning method, the side wall of the polymer film is blocked; hence, the ion injection direction is limited to the top of the polymer film. However, the CYTOP patterning method blocks the top of the polymer film (note that this is different from the CYTOP patterning via orthogonal patterning employed for OECT fabrication in Fig. 4); hence, the ion injection is limited to the side wall of the polymer film. Therefore, four types of devices, that is, DTP-2T device fabricated by parylene patterning, DTP-2T device fabricated by CYTOP patterning, DTP-P device fabricated by parylene patterning, and DTP-P device fabricated by CYTOP patterning methods, can be compared using transient measurements to investigate the effect of ion injection direction relative to molecular orientation. The thickness of DTP-2T and DTP-P films were set to $43 \pm 5$ and $52 \pm 5$ nm, respectively, with relatively small difference, to minimize the possible effect of the side-wall area on ion injection properties.

Figure 5b shows the transient response results of both the DTP-2T and DTP-P devices fabricated through CYTOP patterning. The extracted rise time constant was higher in the case of the DTP-P device, which has an edge-on dominant orientation, showing an opposite tendency compared to the results obtained from the devices fabricated through parylene patterning. The rise time constants of the five devices are plotted in Fig. 5c, Supplementary Figs. 13 and 14, and Supplementary Tables 2 and 3. In addition, the geometry-normalized time constant values were calculated for fair comparison of transient behaviors to eliminate the effect of ion injection area and depth. In case of the devices fabricated through parylene patterning, the area of ion injection is limited to the top of the polymer film, thus, the $\tau$ is proportional to $d(WL)^{1/2}$. In contrast, the CYTOP patterning method blocks the top layer of the polymer film, thus, the $\tau$ is proportional to $[(WL)/2(W+L)]$ $[2d(L+W)]^{1/2}$ (Supplementary Fig. 15), because the time constant is proportional to the product of channel thickness and the square root of nominal channel surface area[28]. The DTP-2T and DTP-P devices fabricated by parylene patterning showed the geometry-normalized $\tau$ values of $9.9 \pm 1.7 \times 10^8$ and $5.4 \pm 0.3 \times 10^8$ s m$^{-2}$, respectively, which suggests the transient response of DTP-P device is faster (Fig. 5f). The DTP-2T and DTP-P devices fabricated by CYTOP patterning showed the geometry-normalized $\tau$ values of $5.7 \pm 0.2 \times 10^8$ and $10 \pm 1.6 \times 10^8$ s m$^{-2}$, respectively (Fig. 5f), revealing that this trend is opposite to those patterned with parylene. Regarding ion injection constrained in two distinct directions, the DTP-P-based OECT exhibits a faster geometry-normalized transient response upon parylene patterning than upon CYTOP patterning whereas the opposite is true for the DTP-2T-based OECT. In the case of the DTP-P devices fabricated by CYTOP patterning, the ion injection direction, and thus, the ion injection efficiency is not strictly opposed by the edge-on dominant molecular faces due to their random in-plane orderings. Nonetheless, it is still obvious that lateral ion injection into DTP-P devices patterned by CYTOP is more frequently impeded than vertical ion injection into those patterned by parylene due to the edge-on-dominant nature without in-plane orderings (Fig. 5a lower panel; more detailed discussion is provided in the next section). Therefore, it can be deduced that the transient response is relatively fast when ions are *vertically* injected into edge-on-dominant films compared with that into face-on-dominant films (Fig. 5a left column), while the transient response is relatively slow when ions are *laterally* injected into edge-on-dominant films compared with that into face-on-dominant films (Fig. 5a right column). We note that these geometry-normalized time constant values in this study suggest that the molecular orientation is one of the important factors influencing time constants. It is, however, not the only factor, considering that paracrystallinity, crystallite size, and molecular weight may affect the apparent capacitance and resistance of the ionic circuit between the channel and the gate electrode and, thereby, the extracted time constant values[43]. Decoupling all these effects on the time constant value would be a future research theme worth exploring.

The further analysis was performed by conducting frequency-response measurements. After obtaining the sine wave $I_D$ by applying a sine wave $V_G$, the $g_m$ values of each device from a certain frequency region were obtained and are plotted in Fig. 5d. Both parylene-patterned devices showed relatively high operational stability in the high-frequency region regardless of the OMIECs. However, the CYTOP-patterned devices showed drastic decrease of ~12% in the $g_m$ values even at 0.2 Hz, which are very slow device operation conditions. For a quantitative comparison of the frequency response of the four devices, the $g_m$ values obtained at 10 Hz instead of 100 Hz are plotted in Fig. 5e because DTP-P device fabricated by CYTOP patterning did not show meaningful $g_m$ values over 10 Hz. The DTP-P device fabricated by parylene patterning showed the highest $g_m$ value of ~148 μS among the four devices, followed by the DTP-2T device fabricated by parylene patterning (~136 μS), DTP-2T device fabricated by CYTOP patterning (~48 μS), and DTP-P device fabricated by CYTOP patterning (~2.7 μS).

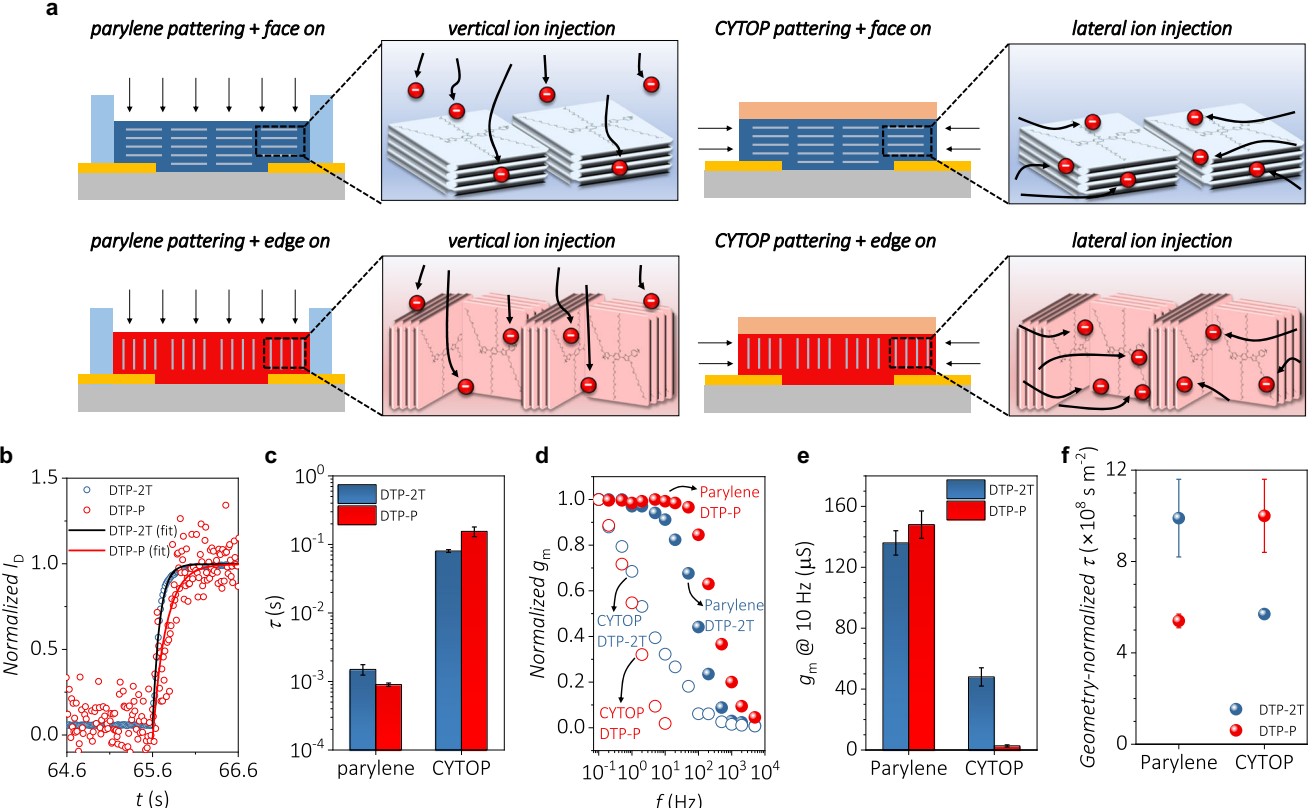

**Fig. 5 | Molecular-orientation-dependent transient behavior of DTP-2T and DTP-P. a** Schematic representation of OECT devices fabricated by different patterning methods. Black arrows indicate ion injection direction when the gate bias is applied to the OECT devices. The active material of the two devices on the top panel (blue) represents a face-on dominant molecular orientation, and that of the two devices on the bottom panel (red) represents an edge-on dominant molecular orientation. **b** Transient response of the drain current at a constant $V_D$ of −0.6 V and a square voltage pulse of 10 s ($V_G$ = −0.8 V) applied at the gate electrode. The devices were fabricated by the CYTOP patterning method. **c** Plot of rise time constant obtained from transient response of DTP-2T- and DTP-P-based OECTs fabricated by parylene and CYTOP patterning methods. The measurements were performed using more than five devices for each set. **d** Plot of frequency-

dependent normalized transconductance obtained from the frequency response measurements of the DTP-2T- and DTP-P-based OECTs fabricated by parylene and CYTOP patterning methods. Biasing condition was $V_D$ = −0.6 V, and sine wave $V_G$ with varying frequency was used. **e** Transconductance obtained from the DTP-2T- and DTP-P-based OECTs by the application of $V_G$ sine wave with a frequency of 10 Hz. Transconductance values were obtained from more than five devices for each set. **f** Geometry-normalized $\tau$ obtained from the DTP-2T and DTP-P-based OECTs fabricated by parylene and CYTOP patterning methods. $\tau$ values were obtained from more than five devices and were normalized based on the channel geometry measurements. The average thickness values, derived from measurements of over five devices, were used for normalization.

Additionally, we compared the geometry-normalized time constants by using the cut-off frequency ($f_c$) from the frequency response results. The DTP-2T and DTP-P devices fabricated by parylene patterning exhibited geometry-normalized $\tau$ values of $8.9 \times 10^7$ and $1.1 \times 10^7$ s m$^{-2}$, respectively, which suggests that the transient response of the DTP-P device is faster than that of DTP-2T device. In the case of CYTOP patterning, DTP-2T and DTP-P devices exhibited geometry-normalized $\tau$ values of $4.2 \times 10^8$ and $5.2 \times 10^8$ s m$^{-2}$, respectively, manifesting that the lateral ion injection is faster in the face-on dominant DTP-2T than the edge-on dominant DTP-P. As demonstrated by the transient measurements, the doping efficiency is higher when the ion injection direction is in parallel with the lamellar stacking direction [100] (i.e., the higher the doping efficiency, the higher is the $g_m$ value). In addition, one more common feature observed from the two measurements (i.e., transient and frequency responses) is that there are large differences in the obtained parameters between the parylene and CYTOP devices, and these differences are up to an order of magnitude. This can be attributed to the difference in the area of the interface between the electrolyte and polymer film in the two device structures. However, the molecular orientation dependency was observed despite such differences, with consistency for both devices fabricated using different methods, demonstrating that the ion mobility depends on the ion

injection direction relative to the dominant molecular orientation of the polymer.

## Discussion
From the abovementioned results, we were able to discuss how ions drift through the OMIEC films with different molecular orientations. First, we assumed three basic principles for the following discussion. (i) There exist two distinct ion drift behaviors when ions encounter crystallized domains, namely, *infiltration through* and *bypassing around* crystalline domains but, in this research, the overall ion drift through the π−π gap (i.e. in-between two pi-stacked polymer backbones) could be ignored due to the relatively large-size hydrated ions (hydrated Cl⁻ ions: ~7.24 Å, π−π distance: ~3.7 Å)[19,25,44]. (ii) The ions can drift through the lamellar gap (the gap between polymer backbones arranged along the (100) direction) due to the presence of hydrophilic glycol side chains and the large distance of the lamellar gap. (iii) The ion drift is the fastest in amorphous regions of the polymer film compared to that through the lamellar gap owing to the difference in the glycol side-chain density. As shown in Fig. 6a, the ions exist inside the OMIEC films even under zero-bias conditions because of the swelling promoted by the hydrophilic glycol side chains[45]. When a bias is applied, the ions drift along with the bias polarity through the

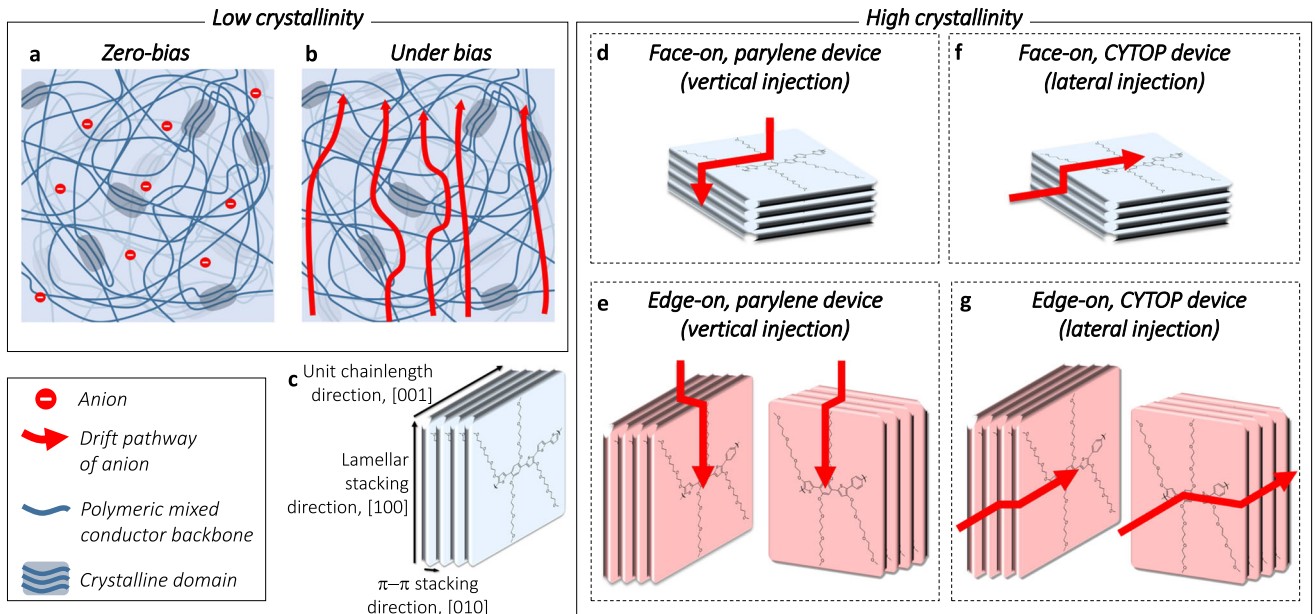

**Fig. 6 | Schematic representations of OMIEC film microstructures and the mechanisms of orientation-dependent transient behavior. a** Amorphous (low crystallinity) OMIEC film swollen by electrolyte. Only anions are displayed for convenience. **b** Amorphous OMIEC film under bias condition. The red arrows represent the drift of anions. **c** Close-up of molecular packing within ordered crystalline domain indicating the repeat directions. Ion-drift behaviors in OMIEC crystalline domains with face-on orientation when the ion injection direction is **d** vertical (parylene patterned) and **e** lateral (CYTOP patterned) to the OMIEC film. Ion-drift behaviors in OMIEC crystalline domains with edge-on orientation when the ion injection direction is **f** vertical (parylene patterned) and **g** lateral (CYTOP patterned) to the OMIEC film.

domain comprising glycol side chains (Fig. 6b). In addition, when the applied bias is sufficiently high to dope the polymer chain, the ions drift to the vicinity of the nearest polymer backbone to compensate for the bias-induced charges, which occurs until the entire film is doped with all electronic charges compensated by ions from the electrolyte. Figure 6b also shows the case of ions interacting with crystalline domains during the drift. Under the bias condition of doping, when the ions face a crystalline domain, the ions would draw close to the surface of the crystalline domain and dope the nearest polymer backbone. Despite ions' infiltration, the drift velocity of ions through crystalline domain is very low because the size of the hydrated ion is larger than π−π stacking distance[19,25,44]. Therefore, if the nearest crystalline domain is fully doped, we assume that the majority of ions move around and dope the next nearest crystalline domain. Because of this bypassing of the doped crystalline domains, the ion-drift pathway becomes longer than that in a purely amorphous region. This dependence of the ion-drift pathway on the polymer microstructure and the existence of the amorphous and crystalline domains are further affected by different molecular orientations of the crystalline domains.

To deal with the different effects of the dominant molecular orientation, we must consider the general structure of crystallized polymeric materials. As shown in Supporting Information (Table S4), the coherence lengths of the (100) and (001) planes are significantly longer than that of the (010) plane. Although the unit chain length of DTP-2T is undiscernible due to the absence of (001) peak, it could be inferred that the coherence length of (001) plane would be similar to or longer than that of (100) plane by considering to the molecular structure and previously reported literatures[13–15,20,37,46–51]. Therefore, when the ion injection direction is in parallel with the π−π stacking direction [010], the *bypassing* pathway of ions becomes significantly longer than when the ion injection direction is in parallel with the lamellar stacking [100] or unit chain length [001] direction (Supplementary Fig. 16 and Fig. 6d–g, see Fig. 6c for clear schematic representation of ion injection direction). In the case of the DTP-P device

patterned with CYTOP (Fig. 6g), two possibilities of ion injection exist due to the absence of in-plane ordering. However, the chance of ion injection occurring in parallel with the π−π stacking direction (Fig. 6g right) is much higher than that of ion injection occurring in parallel with the unit chain length direction (Supplementary Fig. 17). Therefore, we can conclude that in the case of the DTP-P device patterned with CYTOP, the dominant ion injection direction is in parallel with the π−π stacking direction [010]. This indicates that a longer ion drift pathway results in a slower transient response of OECTs (Fig. 5f). Even if the lamellar gap, which can be considered as a shortcut for ion drift, is considered, the local ion mobility in the lamellar gap is lower than that of the amorphous region. We expect that these trends will also be observed in various organic mixed conductors with glycol side chains with decent crystallinity and, therefore, our findings could be utilized as a reference when investigating the molecular-orientation-transient behavior correlation of mixed conductors or for establishing new design rules for organic mixed conductors to carefully modulate material properties for desired applications.

In summary, two DTP-based polymeric mixed conductors were synthesized through direct arylation polymerization. Utilizing two different co-monomer units, that is, phenylene and bithiophene, to manipulate the degree of polymer backbone planarity allowed us to control the dominant molecular orientation of the conducting polymer backbone within crystalline domains relative to the substrate. After investigating the optical and electrochemical properties of DTP-2T and DTP-P, we found that the polymers have similar electrochemical performance in terms of OECT operation. This revealed that the ionic/electronic transducing efficiencies are very similar despite the difference in molecular orientation observed for the two systems. In addition, we analyzed the effect of the molecular orientation of each polymer on the ionic mobility and transient behavior of OECT devices by fabricating the desired experimental platforms, including moving front experiments, data acquisition setup, and function generator. Finally, we demonstrated that the ion injection direction relative to the molecular orientation of the polymer affects the length of the ion-drift

pathway and OECT transient behavior through a new device patterning method (i.e., orthogonal patterning) to limit the ion injection direction and compare the transient and frequency responses of the devices. We believe that this study demonstrates the dependency of ion injection direction relative to molecular orientation exists in term of ion drift in (semi)crystalline polymeric mixed conductors. Moreover, the knowledge accumulated from this research can be a valuable reference for future designs of organic mixed conductor materials, controlling the molecular orientation of existing systems through optimization of processing conditions, and designing new bioelectronic devices benefiting from geometry-optimized configurations.

## Methods

### Chemicals
Chemicals were purchased from Acros Organics, Fluorochem, and TCI and were used without further purification. Solvents were HPLC grade and were purchased from Honeywell and used without further purification.

### Materials characterization
$^1$H and $^{13}$C NMR spectroscopy were carried out on a Bruker AV400 or Bruker AVIII400 spectrometers. Chemical shifts ($\delta$) are quoted in ppm relative to the residual solvent peak. d-CHCl$_3$ was purchased from Cambridge isotopes and d$_2$-TCE was purchased from Fluorochem. HRMS was carried out on Synapt G2-Si High Definition Mass Spectrometer.

### Computational details
All the compounds presented in the electronic structures were optimized via B3LYP 6−311G(d,p) basis set.

### Optical spectroscopy
UV-VIS spectroscopy and spectro-electrochemistry were carried out using a Shimadzu UV3600 UV-vis-nIR spectrometer. Thin films for UV-VIS were prepared using Laurell Technologies WS-650Mz-23NPPB spin coater.

### Electrochemical analysis
Cyclic voltammetry was carried out using a PalmSens 3 electrochemical cell on glassy carbon electrode with Ag/Ag$^+$ reference electrode and platinum wire counter electrode.

### Gel permeation chromatography
GPC was carried out using Shimadzu Prominence GPC system, comprised of a SIL-20A auto sampler, LC-20AT liquid chromatograph, CTO-20A column oven, RID-20A refractive index detector and an SPD-20A UV-Vis detector. HPLC grade chloroform stabilized with amylene was purchased from Acros Organics. Analysis was carried out on Shimadzu's LabSolutions software.

### X-ray diffraction
For XRD spectra, the polymer films prepared on Si substrates were employed. The XRD spectra were collected using an X-ray diffractometer (RigakuD/max-2500) with Cu K$\alpha$ radiation ($\lambda = 1.54$ c5) at 40 kV and 100 mA. Exposure time was set as 1 s and the background peaks were subtracted.

### Grazing incidence wide angle X-ray diffraction
GIWAXD measurements were conducted at the 9 A U-SAXS beamline of the Pohang Light Source (PLS), Republic of Korea. All samples for GIWAXD were prepared by drop-casting of polymer solution (5 mg mL$^{-1}$) on p-Si$^{++}$/SiO$_2$ (300 nm) substrates. The substrate size was 15 mm × 15 mm while the thickness of polymer film was set at ~50 nm. The wavelength of X-rays was 1.12370 Å ($E = 11.025$ keV), and the incidence angle of the beam light was ~0.1°. The size of the incident beam

was 60 μm × 298 μm (vertical and horizontal) at detector position, and the divergence of the beam was 25/45 urad (vertical and horizontal). Typical exposure time was ~12 s, and all measurements were conducted in ambient conditions. The images from GIWAXD were obtained with a 2D FT-CCD module of Rayonix MX170-HS, while the sample-to-detector distance was adjusted to be 225 mm.

### Electrochemical impedance spectroscopy
For EIS measurement, all polymer films were prepared on indium tin oxide (ITO)-coated glass substrates as working electrodes, while the electrode surface other than the active electrode area was passivated using epoxy glue and in contact with an electrolyte solution of 0.1 M NaCl in water. Electrochemical impedance spectra were obtained using PGSTAT304N (Metrohm) equipped with a conventional three-electrode system composed of a working electrode, an Ag/AgCl reference electrode, and a Pt counter electrode, at the frequency range between 0.1 and 10$^5$ Hz with a single sinusoidal signal of $E_{ac} = 10$ mV.

### Ion mobility measurement
Moving front experiment was conducted by fabricating devices as follows. Glass substrates were cleaned by ultrasonication, followed by the electrode pattern deposition [Cr (5 nm)/Au (40 nm)] via conventional photolithography. After polymer films were prepared on the glass substrate with patterned contact electrode, CYTOP layer was fabricated on the polymer film by spin-cast to protect polymer layer from photoresist patterning process. A positive photoresist pattern was prepared on the polymer film with a channel length of 15 mm and a width of 1.5 mm. The non-channel section which are not passivated with the positive photoresist pattern was removed by dry etching. After the removal of residual photoresist, GXR-601 photoresist was spin-coated on top of the polymer film to serve as an ion injection barrier. The whole device areas except contact pads and the open region for electrolyte/active layer junction were passivated with the GXR601 layer. The contact pad was connected to the ground terminal of the source meter and passivated by insulating epoxy (F-301, Alteco), and open region for electrolyte contact was covered with the electrolyte solution of 0.1 M NaCl in water which is in contact with suspended Ag/AgCl electrode wire with a source meter unit. While the dedoping of channel was in progress by applying an appropriate positive bias to the Ag/AgCl electrode, the moving front was monitored using a digital camera (Iphone 8, Apple).

### Reporting summary
Further information on research design is available in the Nature Portfolio Reporting Summary linked to this article.

## Data availability
All data generated in this study are available in the main text or the Supplementary Materials. Source data are provided with this paper.

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

## Acknowledgements

C.B.N. and M.-H.Y. acknowledge support from the Medical Research Council and the Korea Health Industry Development Institute (UK-Korean Partnering Award MC_PC_18066). C.B.N. acknowledges the European Commission for financial support through the MITICS H2020-EU-FET Open project (No. 964677). M.-H.Y. acknowledges a National Research Foundation (NRF) grant funded by the Korean government (MSIT) (NRF-2018M3A7B4070988, NRF-2021R1A2C1013015, NRF-2020M3D1A1030660, NRF-2020M1A2A2080748, and NRF-2017K1A1A2013153).

## Author contributions

J.H.K. and R.H. conceived and designed the experiments. J.H.K., I.L., and S.P. prepared the devices. Data acquisition/analysis was performed by R.H. and P.A.G.F. (NMR, GPC, UV-vis, cyclic voltammetry, spectroelectrochemistry, and computational simulation); J.H.K. and I.-Y.J., (moving front experiment, OECT device characterization); and H.A. (GIWAXD); C.B.N. and M.-H.Y. coordinated the project. The manuscript was written by J.H.K. and R.H. with input from all the authors and revised by C.B.N. and M.-H.Y. All authors contributed to the manuscript and have given approval to the final version of the manuscript.

## Competing interests

The authors declare no competing interests.
