## [Peer Review File · Nature Communications]

REVIEWER COMMENTS

Reviewer #1 (Remarks to the Author):

Overall, I found this to be a very interesting paper that describes the effects of molecular orientation on ion injection in organic electrochemical transistors (OECTs). The work has been very carefully carried out and data carefully considered. Overall, I believe that the work will be impactful to the rest of the community. However, I do think that the paper is not ready for publication and some major issues need to be addressed.

I did have a couple of major questions, though.

1. In the GIWAX section, the authors conclude that DTP-2T shows generally face-on orientation whereas DTP-P shows generally edge-on orientation. They also note that DTP-P is significantly tilted with a tilt angle of 37.5. That is a considerable tilt and so figures 5 and 6 do not accurately represent the molecular orientation of the films. How does the data interpretation change if they consider this tilt more carefully?

2. It is known that some polymers change their molecular orientation upon doping. Did the authors check whether or not these polymers show a change in orientation when the electrolyte is present? This would also change the interpretation of their data.

Just a couple of minor comments:

Line 114: Please indicate the dispersity of the polymers too. Also, average molecular weights are best represented using $(k)g\text{mol}^{-1}$ as units not Da.

Figure 6g. I assume the z direction is the thickness of the film. If so, figure 6g does not show an edge-on polymer – the polymer backbone should still be parallel to the substrate. If so, Figure 6g would look very similar to 6e and I'm not sure that I see a difference in the drift pathways.

Reviewer #2 (Remarks to the Author):

The study investigates OECT transient behavior using two polymers with different film structures and molecular orientations, and patterning techniques. The authors note that most investigative studies focus on steady state performance and not enough on the transient characteristics of materials and devices, which they aim to rectify in this study, using a comprehensive combination of electrochemical, structural, optical techniques. The study successfully demonstrates the dependency of ion injection

direction in relation to molecular orientation as well as the consequent transient device behavior. It is well written and analyzed work.

However, the study does have some inconsistencies and deficiencies in the discussion that need to be addressed, as listed in the comments below. Once those are addressed, the study should be ready for publication.

- Page 6, line 119: "The phenylene comonomer unit causes a significant backbone twist owing to steric crowding from the phenyl hydrogens interfering with the adjacent glycolated thiophene units." It might be pertinent to disclaim this in the abstract and introduction, that the two different polymers differ not only in orientation, but backbone planarity.
- Page 8, line 167: "Ionization potentials (IPs) were estimated from the onset of oxidation." How exactly was this done? Were the CVs done in acetonitrile or aqueous media?
- Page 8, line 171: "This is possibly attributed to the fact that the proportion of electroactive polymer backbone (relative to the side chains) is higher in DTP-2T than in DTP-P which can be rationalized by considering one polymer repeat unit with four triethylene glycol side chain across five aromatic rings in DTP-2T (averaging 0.8 side chain per aromatic unit) and four aromatic units in DTP-P (1 side chain per aromatic unit). Furthermore, the electron-rich thiophene group is beneficial for charge storage when ions compensate for the charges of the polymer backbone." How would the currents compare if they were normalized to the electroactive proportion of the polymer?
- Page 10, line 202: "The μC^* estimated from the orthogonally patterned devices was comparable to that of the parylene patterned devices (Fig. S5)." Was μC^* extracted from g_m vs $[Wd/L][V_{th}-V_g]$ slope? If so, how is μC^* of DTP-P quoted higher than DTP-2T in the parylene patterning method, when the slope is shallower? In which case it would also suggest the different patterning techniques give opposite trends for μC^* of the two polymers.
- Page 10, line 216: "...channel current modulation efficiency..." g_m is the rate of change of IDS with VGS, not "efficiency", which would suggest there is a maximum cap of $dIDS/dVGS$ when "efficiency" is 100%.
- Page 11, line 225 (equation 1): This should be $V_G - V_{Th}$, not the other way around.
- Page 11, line 235: "...by dividing the capacitance value by the nominal volume of the polymer film..." What is nominal volume? How was it obtained?
- Page 12, line 241: "Although the μC^* values of these two polymers are similar and the OECT mobilities are of the same order of magnitude, it can be concluded that the performances of these two polymers are similar, but their molecular orientations are noticeably different." The conclusion is lacking. Why are the μC^* values and the device performances similar? Is there no effect of molecular orientation? Needs further discussion.
- Page 13, line 266: "This clearly indicates that the face-on orientation is highly advantageous for ion drift in lateral direction throughout the mixed conducting polymer film compared to the edge-on

orientation.” Does this mean that edge-on orientation should be advantageous for vertical ion drift? If so, why is this not evident in CVs and spectroelectrochemistry?

- Page 13, line 274: “As shown in the middle panel of Fig. 4f, the ions can be injected through the whole surface of active layer except the bottom surface, which is substrate–active layer interface.” Is there not a passivation layer surrounding the channel? Why are its "sides" exposed? If there is not a passivation layer, does that mean the gold contacts are exposed to electrolyte? Would that not cause for there to be significant parasitic capacitance in the channel?
- Page 16, line 335: “Fig. 5b shows the transient response results of both the DTP-2T and DTP-P devices fabricated through CYTOP patterning.” How comparable are the film thicknesses? If these are not similar enough, then the ion injection surface area will vary and affect response characteristics.
- Page 16, line 351: “Therefore, it can be deduced that the transient response is relatively fast when the ion injection direction is parallel to the polymer backbone orientation, while the transient response is relatively slow when the ion injection direction is not parallel to the polymer backbone orientation.” Will it not be the case that with an edge-on film in CYTOP fabrication (i.e. DTP-P-CYTOP device), there will be two walls/surfaces with perpendicular backbone-ion entry and two with parallel backbone-ion entry? If so, it is no longer strictly only "perpendicular" ion entry, such as in the case of DTP-2T-parylene device.
- Page 17, line 364: “...highest gm value of $\sim 148 \mu\text{S}$ among the four devices, followed by the DTP-2T device fabricated by parylene patterning ($\sim 136 \mu\text{S}$), DTP-2T device fabricated by CYTOP patterning ($\sim 48 \mu\text{S}$), and DTP-P device fabricated by CYTOP patterning ($\sim 2.7 \mu\text{S}$).” These should be reported as geometry normalized values
- Page 17, line 373: “...manifesting that the lateral ion injection relative to polymer orientation is faster.” It is faster specifically when comparing DTP-2T to DTP-P, not overall.

Minor comments:

- Page 14, line 289: [28] is the wrong ref.
- Page 19, line 415: “In general, the unit cell parameters of typical mixed conducting polymers are the longest along the (100) direction, followed by the (001) and (010) directions (Fig. 6c).” It is very hard to distinguish the polymer chemical structure drawn in the unit cell in fig 6c, making it difficult to tell which orientation direction of the backbone is being discussed (for those who are not crystallography experts).

Reviewer #3 (Remarks to the Author):

Review

Ji Hwan Kim et al, “Peculiar Transient Behaviors of OECT Governed by Ion Injection Directionality”

Authors present a study of two closely related organic mixed ionic electronic conductors (OMIEC), denoted DTP-P and DTP-2T. The authors present results from experiments demonstrating that the direction of ionic injection relative to the molecular orientation within the materials affects the performance. Finally, as an important control to rule out other factors, the authors perform tests on samples prepared with different masking schemes to limit the ionic injection direction and demonstrate that the effect persists. The authors present a novel approach to studying OMIECs that sheds light on a previously unexplored aspect of their function and would be of interest to the readers of Nature Communications. However, the authors first need to make some significant revisions to address a lack of clarity in the discussion of the molecular orientation, and should then reconsider their interpretation and conclusions. Afterwards, the revised manuscript may be suitable for publication in Nature Communications. Below are detailed comments.

The authors prepare thin film transistors with active layers made from the two materials and then apply masking schemes in a new way that limits the direction of ionic injection into the active layers to be either in the plane of the layer (lateral) or out of the plane (vertical). The premise of the article is that the two materials are closely matched in their makeup except for differences in the predominant molecular orientation within the thin films, and that this accounts for the differences in the transient and high-frequency performance of the two materials depending on the ionic current injection direction. Moving front experiments also support this conclusion. All of this places a lot of emphasis on the structural analysis of the materials. The structural evidence comes from X-ray diffraction (XRD) and GIWAXS data presented in Figure 2. The authors need to provide more details of these measurements and to carry out more careful analysis of the data to support their conclusions. Equally importantly, the authors should use more precise language to state their thesis.

P 6-7 discussion of material structure

Lines 115-154

This section needs revision. See also comments below for Figure S2

The authors' analysis of the GIWAXS data is mostly confined to a visual inspection of the 2D data and classification of the two materials' predominant orientation as either "face on" or "edge on". These two descriptions refer to the orientation of the conjugated rings. When the scattering feature from the pi-pi stacking is in the out-of-plane direction (q_z), the rings are said to be oriented "face on" (or "plane on"). When the pi-pi stacking feature is oriented along the in-plane direction (q_y), the conjugated rings are said to be oriented "edge on". The naming of these two orientation states comes from the conjugated rings, but they also correspond to orientation of the lamellar scattering, arising from the average separation between the polymer backbones as spaced by the side chains. In a face-on system, the lamellar stacking produces scattering features in the in-plane direction while edge-on systems show lamellar scattering in the out-of-plane direction. (DeLongchamp 2011).

In Figure 2b, the orientation of the DTP-2T material does match the classic description of a face-on system. However the first order lamellar peak, expected to have a maximum in the in-plane direction, is mislabeled as (001), corresponding to the repeating unit separation along the backbone, when it should be labeled (100). The very faint feature labeled (300) can be disregarded.

In Figure 2c, the orientation of the DTP-P material is less clear and appears to have a mixture of face-on orientation, with the pi-pi stacking (010) feature strongest in the out-of-plane direction, and edge-on with the lamellar stacking (n00) strongest in the out-of-plane direction.

In both cases, construction of the pole figure for the (100) and (010) features would give a more quantitative description of the molecular orientation (Savikhin 2018). Integration of the pole figure over different regions of the polar angle χ in the q_z vs q_r plane could be a way to quantify the fractions of the material that are in the face on vs edge on orientations, which could be useful for normalizing the geometrically normalized time constants further (Figure 5 discussion below).

P12 line 240 “This could be attributed to the edge-on orientation of DTP-P, which is favorable for lateral charge transport”

As discussed above, DTP-P is not purely edge-on but has a mixture of orientations.

P13 line 266 “This clearly indicates that the face-on orientation is highly advantageous for ion drift in lateral direction throughout the mixed conducting polymer film compared to the edge-on orientation.”

Note that the lateral direction is parallel to the lamellar stacking direction (100) for the face-on orientation.

P 14 line 300 “This reveals that the ion injection is significantly faster when the ion injection direction is parallel to the ‘face’ of the polymer backbone (i.e., parallel to the lamellar stacking direction). “

The direction “parallel to the lamellar stacking direction” [(100)] is unambiguous. It is in-plane (lateral) for the predominantly face on DTP-2T and predominantly out-of-plane (vertical) for the mixed DTP-P. The direction “parallel to the ‘face’ of the polymer backbone” is unclear. The direction of any Bragg reflection is perpendicular to the lattice planes that cause the reflection. If these lattice planes are what is being referred to as the ‘face’ of the polymer backbone, then more explanation is needed to make this clear. Note that these planes contain the polymer backbone and also the sidechains, and that the lamellar stacking direction is perpendicular to the length of the polymer backbone. [Verploegen 2010]

P 15 line 315 “Although DTP-2T and DTP-P showed that ionic mobilities and frequency-dependent transient behaviors of OECTs depend on the direction of ion injection, there exists still the possibility of material-dependent characteristics (e.g., molecular weight, relative crystallinity) regardless of the molecular orientation of OMIECs. Finally, different patterning methods were employed to demonstrate the molecular orientation dependency of the transient behavior.”

This is a great control experiment to finesse the details of the molecular orientation....

P 16 Line 347 “The DTP-2T and DTP-P devices fabricated by parylene patterning showed the geometry-normalized τ values of 9.9×10^8 and 5.4×10^8 s/m², respectively, which suggests the transient response of DTP-P device is faster. The DTP-2T and DTP-P devices fabricated by CYTOP patterning showed the values of 5.7×10^8 and 1.0×10^9 s/m², respectively, revealing that this trend is opposite to those patterned with parylene.”

However, the interpretation still relies on comparisons of performance of the 2 different materials. Isn't it more interesting to make performance comparisons of the same material with the ion injection constrained in 2 different directions? Should these 4 values (with appropriate error bars) for the geometrically-normalized time constants be highlighted in a figure? Something similar to 5c or 5e, but grouped differently. There would be 2 adjacent bars for $\tau_{\text{norm}}(\text{DTP-2T, parylene})$ and $\tau_{\text{norm}}(\text{DTP-2T, CYTOP})$ and also 2 adjacent bars for $\tau_{\text{norm}}(\text{DTP-P, parylene})$ and $\tau_{\text{norm}}(\text{DTP-P, CYTOP})$. Above the first pair could be an illustration showing the lamellar stacking direction for (face-on) DTP-2T (in-plane or horizontal) and above each bar could be an indication of the ionic injection direction. And similar for the second pair, but for edge-on DTP-P with the vertical lamellar stacking.

Another question: No error bars are given for the geometrically normalized time constants, but it looks like the two fastest time constants and the two slowest time constants are almost the same. Is this result significant? Or just a coincidence? I'm asking because besides the geometry, the other normalization that is needed is to normalize with respect to the molecular orientation. DTP-2T seems strongly face on while DTP-P is more mixed. If the normalization also took into account the mean orientation, would the time constants agree quantitatively?

P 16 line 351 “ Therefore, it can be deduced that the transient response is relatively fast when the ion injection direction is parallel to the polymer backbone orientation, while the transient response is relatively slow when the ion injection direction is not parallel to the polymer backbone orientation.”

Authors should use more precise language here. The direction along the polymer backbone should be the (001) direction.

I believe the data presented in Figure 5, together with correct interpretation of the structural data in Figure 2 means that the opposite is true. The transient response is relatively fast when the ion injection direction is parallel to the lamellar stacking direction, which means when the ion injection is perpendicular to the polymer backbone. Equivalently, the transient response is relatively fast when the ion injection direction is perpendicular to the pi-pi stacking direction.

P 18 line 392 “ii) the ions can drift through the lamellar gap (the gap between polymer backbones arranged along the (100) direction) due to the presence of hydrophilic glycol side chains and the large distance of the lamellar gap;”

As discussed above, the evidence suggests ionic injection is faster when it is parallel to the lamellar stacking direction. The lamellar stacking direction is perpendicular to the lamellae. Moving in the (100) direction means moving along the glycol side chains.

P18 line 389 “(i) ions may infiltrate into the gap of hydrophobic π - π stacks but their drift velocity is very low due to the relatively large solvation shells of hydrated ions, thus the effect of the overall ion drift through the crystalline polymer domain could be ignored^{19,25,43}. due to the relatively large hydrated ions,”

In contrast to the ‘lamellar gaps’ mentioned above, the ions do move more quickly when they move perpendicular to the pi-pi stacking direction, so they do move more rapidly when they encounter the ‘pi-pi gaps’.

P 19 Line 406

“the drift velocity of ions through crystalline domain is very low because the size of the hydrated ion is larger than π - π stacking distance”

It would be helpful for the authors to make the comparison of these sizes explicitly in the text.

P 19 418-420 “Assuming that there exists densely packed polymer crystal consisting of many unit cells, and that the number of unit cell along with each axis ([100], [010]) is identical regardless of axis, the coherence lengths of (100) and (010) planes will be proportional to the length of each unit cell parameter”

Instead of making this assumption, authors should analyze the width of the (010) and (100) peaks and obtain the mean number of unit cells contributing to the diffraction, according to the Debye-Scherrer theory, taking into account the experimental broadening from the use of area detectors [Smilgies 2009]

P 23 GIWAXD methods section should describe the experiment more fully.

Authors should provide a reference describing the beamline and instrument used in this study.

Line 490 “c5” should be the Angstrom symbol.

Authors should provide the dimensions of the samples, the size of the incident beam, the divergence of the beam. These parameters contribute toward evaluating the experimental resolution. Authors should also report typical exposure times, and the environment of the sample during the measurement (vacuum? He? Ambient?). The authors should also define the regions of integration used for the linecuts presented in Figures 2d, e.

P 30 Figure 2

Figure 2a. (1) The pseudocolor scale bars in the GIWAXD data show a logarithmic scale that makes the lamellar diffraction peaks ($n00$) for the DTP-P material (2c) plainly visible up to 3rd order. However, the XRD data appear to be on a linear scale and only the first order lamellar peak is visible. If the data were on a log scale, would the higher order peaks also be visible? Or is the data too noisy to show the higher order peaks? Perhaps longer integration times would reveal these features. No details about the x-ray exposure times or background subtraction are given in the experimental description on p. 23. (2) The intensity is plotted as a function of the scattering angle, 2θ . It would be more meaningful to convert this to the equivalent $q_z = (4\pi/\lambda)\sin(\theta)$, which would allow more direct comparison with the GIWAXD data. (3)

Figure 2, c. 2D data should be reshaped in the representation q_z vs $q_r = \sqrt{q_x^2 + q_y^2}$, which would make it clear that a wedge shaped region is not observed in the GIWAXD measurement [Rivnay 2012].

P 34 Figure 5

A. Presents schematics of the 2 different mask patterning approaches applied to idealized face on and edge on materials. (1) Judging from the schematic of the OMIEC unit cell in Figure 6C, which shows the (100) direction as the long direction, the edge on material is not illustrated correctly. For edge on material, the (100) direction is vertical, so the red tiles in the bottom panels of columns 2 and 4 should have their long dimensions oriented vertically (2) The labels on the 2nd and 4th columns are inconsistent. "Vertical" refers to the lab/substrate frame of reference. For the Parylene patterning (second column), the ion injection is always vertical. For the CYTOP patterning (4th column), the ion injection is always "lateral" (or horizontal). So the label in the fourth column, second row is incorrect. On the other hand, "parallel" refers to a comparison of two vectors, one of which is the ion injection direction. The other is not clear. From the illustration, in the 4th column top panel, "parallel injection" means injection parallel to the lamellar stacking (100) direction, which is correct. Meanwhile, as illustrated in the 2nd column bottom panel, "parallel injection" means parallel to the backbone repeat direction (001) which is incorrect. The red tiles should be oriented so their long dimension is vertical, then the (100) direction will be (anti)parallel to the injection.

Supporting Material, p 14, Figure S2

The 2D data shows a maximum in the feature at $q \sim 1.5 \text{ \AA}^{-1}$ not in the in-plane or out-of-plane directions, but at 37.5 deg measured from the in-plane direction. The diagram below the data is intended to aid in interpreting the origin of this feature. However the interpretation presented (see main text, p 7) confuses concepts to arrive at an erroneous interpretation. Consideration of the angle of the maximum in the scattering data to obtain a molecular tilt angle as in Figure S2 comes from 2D crystallography of amphiphilic monolayers, as described by [Kaganer 1999], relies on the intersection of the reciprocal disk representing the form factor of rod-like molecules and the six Bragg rods in the reciprocal lattice representing a hexagonally packed 2D crystal when the rods are tilted (see Figure 5 in [Kaganer 99] and discussion in text). Following the authors' logic, this feature would not correspond to the lamellar spacing, but would correspond to the mean spacing between the ethylene-glycol chains.

However, this cannot be the case because only 1 such feature is observed. As stated in [Kaganer 99], the tilting of the rods breaks the symmetry, and so additional peaks due to the crystalline EG chains should also be observed at different q values. Instead, this feature appears to be due the broad distribution of the π - π stacking in the material.

References

Kaganer, Möhwald and Dutta, "Structure and phase transitions in Langmuir monolayers", *Reviews of Modern Physics*, Vol. 71, No. 3, pp 799-819 (April 1999)

Rivnay, Mannsfeld, Salleo and Toney, *Chem. Rev.* 112 5488-5519 (2012)

Savikhin, Babics, Toney et al, "Impact of Polymer Side Chain Modification on OPV Morphology and Performance," *Chemistry of Materials* 30, 21, 7872-7884 (2018) [DOI: 10.1021/acs.chemmater.8b03455]

Smilgies, "Scherrer grain-size analysis adapted to grazing incidence scattering with area detectors, *J Appl Cryst* 42, 1030-1034 (2009) [doi:10.1107/S0021889809040126]

Verploegen, Modal, Bettinger, Sok, Toney and Bao, " Effects of Thermal Annealing Upon the Morphology of Polymer–Fullerene Blends," *Adv. Funct. Mater.* 20, 3519-3529 (2010) [https://doi.org/10.1002/adfm.201000975]

Responses to Reviewer #1's comments

Overall, I found this to be a very interesting paper that describes the effects of molecular orientation on ion injection in organic electrochemical transistors (OECTs). The work has been very carefully carried out and data carefully considered. Overall, I believe that the work will be impactful to the rest of the community. However, I do think that the paper is not ready for publication and some major issues need to be addressed.

We are grateful for Reviewer #1's appreciation of our research. We made our best to revise the manuscript in response to his/her comments.

[Comment #1]

I did have a couple of major questions, though.

1. In the GIWAX section, the authors conclude that DTP-2T shows generally face-on orientation whereas DTP-P shows generally edge-on orientation. They also note that DTP-P is significantly tilted with a tilt angle of 37.5. That is a considerable tilt and so figures 5 and 6 do not accurately represent the molecular orientation of the films. How does the data interpretation change if they consider this tilt more carefully?

[Response #1]

We thank Reviewer #1 for his/her constructive comments. During the revision process, Reviewer #3 also pointed out in comment #17 regarding the tilting of the molecules and suggested that it should be corrected according to the previous literature that thoroughly addressed the molecular tilting angle (*Rev. Mod. Phys.* **71**, 779 (1999) 'Structure and phase transitions in Langmuir monolayers'). In this article, the authors described molecular tilting by using 2D crystallography of amphiphilic monolayers and relied on the intersection of the reciprocal lattice representing a hexagonally packed 2D crystal, particularly, in the case of tilted rods (see Figure 5 of this article and discussion in text). According to this, the tilting of the rod structure breaks the symmetry and, thereby, additional peaks are supposed to be observed at different q values due to the existence of crystalline ethylene glycol side-chains. In our research, however, 2D GIWAXD results do not show such features. Therefore, based on further analysis,

we have reached a new conclusion and removed the discussion of tilting angle. We have accordingly revised the crystallographic analysis in the main text, as the ring pattern of DTP-P is attributed to the presence of SiO₂ substrate.

At page 7: Therefore, it is not relevant to the dominant molecular orientation of the polymer film. In the 2D GIWAXD pattern of both DTP-2T and DTP-P, a broad peak corresponding to a spacing of ~ 4.2 Å was observed, indicating the presence of SiO₂ substrate³⁶, revealing the broad distributions of π - π stacking in the film. We also compared the pole figures of two polymers (Fig. S2) and concluded that DTP-P exhibit mixed orientations, but the edge-on feature is rather dominant over the face-on feature.

At page 28: 36. Kraner, S., Koerner, C., Leo, K. Dielectric function of a poly(benzimidazobenzophenanthroline) ladder polymer, *Phys. Rev. B Condens. Matter* **91**, 195202 (2015).

[Comment #2]

2. It is known that some polymers change their molecular orientation upon doping. Did the authors check whether or not these polymers show a change in orientation when the electrolyte is present? This would also change the interpretation of their data.

[Response #2]

We appreciate Reviewer #1's constructive comment. As shown below, the actual *ex-situ* electrochemical GIWAXD patterns and corresponding line-cut profiles indicate that the dominant molecular orientations of DTP-P and DTP-2T on ITO/glass substrates are preserved even after electrochemical doping/dedoping. We added the detailed analysis in the revised text.

Figure R1. 2D GIWAXS patterns of as-cast, oxidized, and restored (reduced back to neutral) DTP-P and DTP-2T films spin-cast onto ITO-coated glass.

Figure R2. Out-of-plane (q_z ; upper row) and in-plane (q_{xy} ; lower row) line-cuts of 2D GIWAXS data obtained from DTP-P and DTP-2T films spin-cast onto ITO-coated glass

substrates; GIWAXS results were collected on the as-cast films (black lines), after electrochemical oxidation (red lines), and after subsequently restored to the neutral state (blue lines).

At page 11: Finally, the OECT mobility was decoupled from the μC^* value, and DTP-P showed higher hole mobility ($0.61 \text{ cm}^2 \text{ V}^{-1} \text{ s}^{-1}$) than DTP-2T ($0.39 \text{ cm}^2 \text{ V}^{-1} \text{ s}^{-1}$) (Fig. S8f). Considering electrochemical doping/dedoping minimally affected the dominant molecular orientations in these polymer films³¹, this trend could be attributed to the edge-on dominant mixed orientations in DTP-P which is still favorable for lateral charge transport compared with predominant face-on orientation^{37,42}.

At page 28: 31. R. Halaksa, J. H. Kim, K. J. Thorley, P. A. Gihooley-Finn, H. Ahn, A. Savva, M. -H. Yoon, and C. B. Nielsen, The influence of regiochemistry on the performance of organic mixed ionic and electronic conductors. *Angew. Chem. Int. Ed.* e202304390 (2023).

[Comment #3]

Line 114: Please indicate the dispersity of the polymers too. Also, average molecular weights are best represented using (k)g mol⁻¹ as units not Da.

[Response #3]

In response to Reviewer #1's suggestion, we revised the text as follows.

At page 6: Thereafter, chloroform was used to isolate the purified materials with weight-average molecular weights of 42 kg mol⁻¹ (dispersity 1.9) and 45 kg mol⁻¹ (dispersity 2.8) for DTP-P and DTP-2T, respectively³¹.

At page 28: 31. R. Halaksa, J. H. Kim, K. J. Thorley, P. A. Gihooley-Finn, H. Ahn, A. Savva, M. -H. Yoon, and C. B. Nielsen, The influence of regiochemistry on the performance of organic mixed ionic and electronic conductors. *Angew. Chem. Int. Ed.* e202304390 (2023).

[Comment #4]

Figure 6g. I assume the z direction is the thickness of the film. If so, figure 6g does not show an edge-on polymer – the polymer backbone should still be parallel to the substrate. If so, Figure 6g would look very similar to 6e and I'm not sure that I see a difference in the drift pathways.

[Response #4]

We thank Reviewer #1 for his/her criticism on the schematic illustration of suggested molecular orientations and corresponding ion drift pathways. For clarification, we fully revised Figure 6 and main text as follows.

At page 20 : Therefore, when the ion injection direction is in parallel with the π - π stacking direction [010], the bypassing pathway of ions becomes significantly longer than when the ion injection direction is in parallel with the lamellar stacking [100] or unit chain length [001] direction (Fig. S14 and Fig. 6d to g, see Figure 6c for clear schematic representation of ion injection direction). In the case of the DTP-P device patterned with CYTOP (Fig. 6g), two possibilities of ion injection exist due to the absence of in-plane ordering. However, the chance of ion injection occurring in parallel with the π - π stacking direction (Fig. 6g right) is much higher than that of ion injection occurring in parallel with the unit chain length direction (Fig. S15). Therefore, we can conclude that in the case of the DTP-P device patterned with CYTOP, the dominant ion injection direction is in parallel with the π - π stacking direction [010]. This indicates that a longer ion drift pathway results in a slower transient response of OECTs (Fig. 5f).

Figure 6. Schematic illustrations of OMIEC film microstructures and the mechanisms of ion-injection-directionality-dependent transient behavior. **a** Amorphous (low crystallinity) OMIEC film swollen by electrolyte. Only anions are displayed for convenience. **b** Amorphous OMIEC film under bias condition. The red arrows represent the drift of anions. **c** Close-up of molecular packing within ordered crystalline domain indicating the crystallographic repeat directions. Ion-drift behaviors in OMIEC crystalline domains with face-on orientation when the ion injection direction is **d** vertical (parylene patterned) and **e** lateral (CYTOP patterned) to the OMIEC film. Ion-drift behaviors in OMIEC crystalline domains with edge-on orientation when the ion injection direction is **f** vertical (parylene patterned) and **g** lateral (CYTOP patterned) to the OMIEC film.

At page 30 of supporting information:

Figure S15. Schematic illustration of molecular packing within edge-on oriented crystalline

domain. The diagram also shows the incident angle range when the ion injection occurs in the DTP-P device patterned with CYTOP.

=====

End of Responses to Reviewer #1's comments

=====

=====
Responses to Reviewer #2's comments
=====

The study investigates OECT transient behavior using two polymers with different film structures and molecular orientations, and patterning techniques. The authors note that most investigative studies focus on steady state performance and not enough on the transient characteristics of materials and devices, which they aim to rectify in this study, using a comprehensive combination of electrochemical, structural, optical techniques. The study successfully demonstrates the dependency of ion injection direction in relation to molecular orientation as well as the consequent transient device behavior. It is well written and analyzed work.

However, the study does have some inconsistencies and deficiencies in the discussion that need to be addressed, as listed in the comments below. Once those are addressed, the study should be ready for publication.

We thank Reviewer #2 for appreciating the value of our research. We did our best to improve the quality of our research by reflecting his/her comments in the revised manuscript.

[Comment #1]

Page 6, line 119: “The phenylene comonomer unit causes a significant backbone twist owing to steric crowding from the phenyl hydrogens interfering with the adjacent glycolated thiophene units.” It might be pertinent to disclaim this in the abstract and introduction, that the two different polymers differ not only in orientation, but backbone planarity.

[Response #1]

We appreciate Reviewer #2's comment. We revised the main text accordingly as follows.

At page 2: Two polymers with similar electrical, ionic, and electrochemical characteristics but distinct backbone planarities and molecular orientations were successfully synthesized by varying the co-monomer unit (2,2'-bithiophene or phenylene) in conjunction with a novel 1,4-dithienylphenylene-based monomer.

At page 4: In this research, two 1,4-dithienylphenylene(DTP)-based polymers with four

triethylene glycol side chains per repeating unit were successfully synthesized to investigate the correlation between backbone planarity-dependent molecular orientation and transient OECT characteristics.

[Comment #2]

Page 8, line 167: “Ionization potentials (IPs) were estimated from the onset of oxidation.” How exactly was this done? Were the CVs done in acetonitrile or aqueous media?

[Response #2]

For clarification, we revised the main text by describing the detailed experimental condition as follows.

At page 8: Ionization potentials (IPs) were estimated from the onset of oxidation in the cyclic voltammograms of polymer films with 0.1 M TBAPF₆ in acetonitrile as supporting electrolytes. DTP-2T has an IP of ~4.20 eV, whereas DTP-P has a higher IP of ~4.44 eV (Fig. 3c and Fig. S3).

At page 15 of the Supporting Information, we have added the cyclic voltammograms with the tangents used to determine the onsets of oxidation:

Figure S3. Cyclic voltammograms of polymer films prepared by using chloroform solution (5 mg mL⁻¹) with a scan rate of 50 mV s⁻¹ versus Ag/Ag⁺ reference electrode (a,b) with 0.1 M

TBAPF₆ as electrolyte in acetonitrile and (c,d) with 0.1 NaCl as electrolyte in water. Note that the red-colored tangents are used to determine the onsets of oxidation.

[Comment #3]

Page 8, line 171: “This is possibly attributed to the fact that the proportion of electroactive polymer backbone (relative to the side chains) is higher in DTP-2T than in DTP-P which can be rationalized by considering one polymer repeat unit with four triethylene glycol side chain across five aromatic rings in DTP-2T (averaging 0.8 side chain per aromatic unit) and four aromatic units in DTP-P (1 side chain per aromatic unit). Furthermore, the electron-rich thiophene group is beneficial for charge storage when ions compensate for the charges of the polymer backbone.” How would the currents compare if they were normalized to the electroactive proportion of the polymer?

[Response #3]

Beside the film microstructure, we admit that the electroactive proportion of a given conjugated polymer (relative to the side chains) may affect the current density observed in the corresponding cyclic voltammogram. Simultaneously, the current density could be also influenced by the electron-richness of the conjugated backbone structure (e.g., thiophene vs. phenylene group). Therefore, it would not be straightforward to extract a meaningful conclusion by comparing current densities which are normalized only by the electroactive proportion of polymer as suggested.

[Comment #4]

Page 10, line 202: “The μC^* estimated from the orthogonally patterned devices was comparable to that of the parylene patterned devices (Fig. S5).” Was $u C^*$ extracted from g_m vs $[Wd/L][V_{th}-V_g]$ slope? If so, how is $u C^*$ of DTP-P quoted higher than DTP-2T in the parylene patterning method, when the slope is shallower? In which case it would also suggest the different patterning techniques give opposite trends for $u C^*$ of the two polymers.

[Response #4]

As Reviewer #2 pointed out, the μC^* values of DTP-P and DTP-2T were extracted from g_m vs.

[Wd/L][V_G-V_{Th}] plots. However, the μC^* values can only be accurately obtained from linear plots following the equation $g_m = \left(\frac{Wd}{L}\right)\mu C^*(V_G - V_{Th})$. The slope of a plot with a logarithmic scale merely indicates the degree of proportionality between g_m and the channel geometry since it shows the correlation between $\log(g_m)$ and $\log([Wd/L][V_G - V_{Th}])$. Therefore, a shallower slope in the DTP-P parylene patterning method does not necessarily imply a lower μC^* value compared to DTP-2T. Figure R3 clearly shows that the slope of the fitted curve in linear plot indicates a higher μC^* value for DTP-P compared to DTP-2T. In response to the second comment, despite some variations, they are still within standard deviations. Therefore, it is hard to conclude that the extracted μC^* values depend on the device fabrication/patterning method.

Figure R3. Logarithmic (top) and linear (bottom) plots of g_m vs. $[Wd/L][V_G - V_{Th}]$ for DTP-P and DTP-2T devices fabricated with parylene patterning methods.

To avoid the possible confusion, we revised the main text as follows.

At page 10: The μC^* values estimated from the orthogonally patterned devices were comparable to those of the parylene-patterned devices while their variations are within the standard deviation (Fig. S6).

[Comment #5]

Page 10, line 216: "...channel current modulation efficiency..." g_m is the rate of change of IDS with VGS, not "efficiency", which would suggest there is a maximum cap of dIDS/dVGS when "efficiency" is 100%.

[Response #5]

We are grateful for Reviewer #2's suggested correction. Accordingly, we revised the sentence as follows.

At page 10: For each polymer, the transconductance (g_m), that is, the rate of change in I_D with V_G , is plotted in Fig. S8c.

[Comment #6]

Page 11, line 225 (equation 1): This should be $V_G - V_{Th}$, not the other way around.

[Response #6]

In response to Reviewer #2's comment, we corrected equation 1 and the corresponding text.

At page 11: For fair comparison, the g_m values should be normalized by a geometrical factor and biasing condition because g_m is obtained from the product of both these factors, as shown in equation 1.

$$g_m = \frac{\partial I_D}{\partial I_G} = \mu C^* \frac{Wd}{L} (V_G - V_{Th}) \quad (1)$$

At page 11: By fabricating OECTs with channel lengths varying from 20 to 80 μm and plotting the corresponding g_m values as a function of $(Wd/L)(V_G - V_{Th})$ (Figs. S8d and S10), it was found that DTP-2T and DTP-P have very similar figure-of-merit (μC^*) values.

[Comment #7]

Page 11, line 235: "...by dividing the capacitance value by the nominal volume of the polymer film..." What is nominal volume? How was it obtained?

[Response #7]

The nominal volume of a given polymer film is obtained from the film thickness (d) measured by surface profiler and the area (A) exposed to the aqueous electrolyte solution during the electrochemical impedance spectroscopy measurement. Therefore, the scheme of electrochemical impedance spectroscopy measurement including the definition of film dimensions was added to the revised supplementary information for better understanding.

At supporting information page 19:

Figure S6. Schematic illustration of electrochemical impedance spectroscopy (EIS) measurement. The nominal volume of the polymer film is obtained from the film thickness (d) measured by surface profiler and the area (A) exposed to the aqueous electrolyte solution during the electrochemical impedance spectroscopy measurement.

[Comment #8]

Page 12, line 241: "Although the μC^* values of these two polymers are similar and the OECT

mobilities are of the same order of magnitude, it can be concluded that the performances of these two polymers are similar, but their molecular orientations are noticeably different.” The conclusion is lacking. Why are the μC^* values and the device performances similar? Is there no effect of molecular orientation? Needs further discussion.

[Response #8]

As Reviewer #2 commented, the molecular orientation in each polymer film influences the corresponding μC^* value. It is well known that the molecular orientation affects charge transport behaviors in a conjugated polymer film and the edge-on orientation relevant to the substrate is favorable for the lateral charge transport (*J. Am. Chem. Soc.* **138**, 10252 (2016) ‘Molecular design of semiconducting polymers for high-performance organic electrochemical transistors’, *ACS Appl. Polym. Mater.* **1**, 1257 (2019) ‘Understanding comparable charge transport between edge-on and face-on polymers in a thiazolothiazole polymer system’, *Adv. Funct. Mater.* **24**, 6270, (2014) ‘Systematic investigation of side-chain branching position effect on electron carrier mobility in conjugated polymers’). Indeed, the mobility value of edge-on dominant DTP-P ($0.61 \text{ cm}^2 \text{ V}^{-1} \text{ s}^{-1}$) is higher than that of face-on dominant DTP-2T ($0.39 \text{ cm}^2 \text{ V}^{-1} \text{ s}^{-1}$). However, the C^* value of DTP-2T (166 F cm^{-3}) is higher than that of DTP-P (113 F cm^{-3}), which can be attributed to the difference in the proportion of electrochemically active backbone core structure. Considering these two factors together, μC^* values of these two polymers are in a very similar range, even though the dominant molecular orientation is different. It is a coincidence that these polymers exhibit similar μC^* values despite their distinct molecular orientations. But, simultaneously, this feature is the reason why these two polymers were selected as the best model system to investigate the effect of molecular orientation on the transient response of OECTs, while the overall figure-of-merits of OECT (i.e., μC^*) remain invariant. For clarification, we revised the main text as follows.

At page 12: Although the μC^* values of these two polymers are similar, the molecular orientations of these two polymers are very different (i.e., edge-on dominant DTP-P vs. face-on dominant DTP-2T), which suggests that these two polymers are an ideal model system to investigate the effect of molecular orientation on the transient response of OECTs while the overall figure-of-merits remain similar.

[Comment #9]

Page 13, line 266: “This clearly indicates that the face-on orientation is highly advantageous for ion drift in lateral direction throughout the mixed conducting polymer film compared to the edge-on orientation.” Does this mean that edge-on orientation should be advantageous for vertical ion drift? If so, why is this not evident in CVs and spectroelectrochemistry?

[Response #9]

As Reviewer #2 pointed out, the edge-on orientation is advantageous for vertical ion drift and, indeed, this was demonstrated using the OECT devices fabricated with different patterning methods to control the ion injection directionality. Nonetheless, the preferential ion drift directionality depending on the dominant molecular orientation appears obvious only in a relatively short time scale, so it could be observed from the transient OECT response (~ ms time scale) but not from the cyclic voltammetry (CV) measurement with much longer time scale and the steady-state spectroelectrochemistry analysis. In response to Reviewer #2’s comment, we revised the main text as follows.

At page 14: The rise time constants for DTP-2T and DTP-P were estimated to be 1.8 and 0.9 ms, respectively. **These results demonstrate that the transient response measurement is effective for examining the ion drift behavior considering that the time scale of ion drift is too fast to be observed by conventional (steady-state) cyclic voltammetry and spectroelectrochemistry.**

[Comment #10]

Page 13, line 274: “As shown in the middle panel of Fig. 4f, the ions can be injected through the whole surface of active layer except the bottom surface, which is substrate–active layer interface.” Is there not a passivation layer surrounding the channel? Why are its "sides" exposed? If there is not a passivation layer, does that mean the gold contacts are exposed to electrolyte? Would that not cause for there to be significant parasitic capacitance in the channel?

[Response #10]

We appreciate Reviewer #2’s comment on the parasitic capacitance. For clarification, note that a schematic illustration of a passivated OECT device is presented below (Fig. R4). First, although gold electrodes are partially exposed to an aqueous electrolyte solution after the

deposition of SU-8 passivation layer as shown in Fig. R5, the actual contact area with electrolytes is almost negligible compared with the whole area of the gold electrodes. In addition, the effective capacitance of the polymer film is typically higher by two orders of magnitude than the actual electrical double-layer capacitance at the interface between metal electrodes and aqueous electrolytes (*Proc. Natl. Acad. Sci. U.S.A.* **113**, 12017 (2016), ‘Controlling the mode of operation of organic transistors through side-chain engineering’). Therefore, we assume that the effect of parasitic capacitance is negligible although gold electrodes were partially exposed to the aqueous electrolyte solution in the present OECT structure.

Figure R4. Schematic illustration of a passivated OECT device.

Figure R5. The optical micrograph of an OECT device after active channel patterning and SU-8 passivation process.

In response to Reviewer #2’s comment, we added the detailed device structure information to Supporting Information as follows.

At page 17 of supporting information: The channel width was 80 μm defined by positive photoresist pattern, and the length of channel was varied from 20 to 80 μm by the contact

electrode pattern. Note that the effective capacitance of the electrical double layer at the interface between metal electrodes and aqueous electrolytes is negligible compared with the effective capacitance of the polymer channel.

[Comment #11]

Page 16, line 335: “Fig. 5b shows the transient response results of both the DTP-2T and DTP-P devices fabricated through CYTOP patterning.” How comparable are the film thicknesses? If these are not similar enough, then the ion injection surface area will vary and affect response characteristics.

[Response #11]

The thicknesses of DTP-2T and DTP-P films patterned with the CYTOP method is 43 ± 5 and 52 ± 5 nm, respectively, and the corresponding time constant effect is extracted. Additionally, for a fair comparison, the geometry-normalized time constant values were calculated just in case of film thickness-dependent effect, and they showed a similar trend of transient response (5.7×10^8 and 1.0×10^9 s m⁻² for DTP-2T and DTP-P, respectively). Accordingly, we revised the main text to add the relevant description on film thickness.

At page 16: ... to investigate the effect of ion injection direction relative to molecular orientation. The thicknesses of DTP-2T and DTP-P films were set to 43 ± 5 and 52 ± 5 nm, respectively, with relatively small difference, to minimize the possible effect of the side-wall area on ion injection properties.

[Comment #12]

Page 16, line 351: “Therefore, it can be deduced that the transient response is relatively fast when the ion injection direction is parallel to the polymer backbone orientation, while the transient response is relatively slow when the ion injection direction is not parallel to the polymer backbone orientation.” Will it not be the case that with an edge-on film in CYTOP fabrication (i.e. DTP-P-CYTOP device), there will be two walls/surfaces with perpendicular backbone-ion entry and two with parallel backbone-ion entry? If so, it is no longer strictly only "perpendicular" ion entry, such as in the case of DTP-2T-parylene device.

[Response #12]

We thank Reviewer #2 for the constructive comment on the ion injection direction relative to the polymer backbone. Since the edge-on dominant DTP-P film does not exhibit any in-plane ordering as shown in GIWAXD pattern, all the walls/surfaces should be randomly oriented for ion entry in the case of DTP-P-CTYOP devices. Therefore, the backbone-ion entry is not limited to solely ‘perpendicular’ as Reviewer #2 pointed out. Nonetheless, the edge-on dominant DTP-P-CTYOP devices with lateral ion injection should show more chances of ‘perpendicular’ backbone-ion entry than face-on dominant DTP-P-parylene devices with vertical ion injection, suggesting that the transient responses of OECTs are significantly affected by the molecular orientation, particularly, with respect to the major ion injection/drift direction. For clarification, we revised the manuscript as follows.

At page 17: The DTP-2T and DTP-P devices fabricated by CYTOP patterning showed the geometry-normalized τ values of $5.7 \pm 0.2 \times 10^8$ and $10 \pm 1.6 \times 10^8$ s/m², respectively (Fig. 5f), revealing that this trend is opposite to those patterned with parylene. Regarding ion injection constrained in two distinct directions, the DTP-P-based OECT exhibits a faster geometry-normalized transient response upon parylene patterning than upon CYTOP patterning whereas the opposite is true for the DTP-2T-based OECT. In the case of the DTP-P devices fabricated by CYTOP patterning, the ion injection direction, and thus, the ion injection efficiency is not strictly opposed by the edge-on dominant molecular faces due to their random in-plane orderings. Nonetheless, it is still obvious that lateral ion injection into DTP-P devices patterned by CYTOP is more frequently impeded than vertical ion injection into those patterned by parylene due to the edge-on-dominant nature without in-plane orderings (Figure 5a lower panel; More detailed discussion is provided in the next section). Therefore, it can be deduced that the transient response is relatively fast when ions are *vertically* injected into edge-on-dominant films compared with that into face-on-dominant films (Fig. 5a left column), while the transient response is relatively slow when ions are *laterally* injected into edge-on-dominant films compared with that into face-on-dominant films (Fig. 5a right column).

[Comment #13]

Page 17, line 364: “...highest gm value of ~148 μ S among the four devices, followed by the DTP-2T device fabricated by parylene patterning (~136 μ S), DTP-2T device fabricated by

CYTOP patterning ($\sim 48 \mu\text{S}$), and DTP-P device fabricated by CYTOP patterning ($\sim 2.7 \mu\text{S}$).” These should be reported as geometry normalized values.

[Response #13]

We appreciate Reviewer #2’s constructive comments. The goal of the frequency response measurement is to test the device’s response rate by applying a sine wave V_G signal. If the device is fast enough to synchronize with the V_G signal, the g_m value of the device will be maintained. However, if it is not fast enough, the g_m value will decrease, in such cases, the ions may not fully follow the V_G signal, leading to incomplete doping/dedoping of the channel at that frequency of sine wave signal. Therefore, the g_m value obtained at a certain frequency indicates how well the device can follow fast-frequency signals. For a fair comparison, these values were converted to time constant value (τ) using the cut-off frequency obtained from frequency measurements, and they were also normalized by the geometry of channel.

[Comment #14]

Page 17, line 373: “...manifesting that the lateral ion injection relative to polymer orientation is faster.” It is faster specifically when comparing DTP-2T to DTP-P, not overall.

[Response #14]

We appreciate Reviewer #2’s constructive comment. Accordingly, we revised the main text as follows.

At page 18 : In the case of CYTOP patterning, DTP-2T and DTP-P devices exhibited geometry-normalized τ values of 4.2×10^8 and $5.2 \times 10^8 \text{ s m}^{-2}$, respectively, manifesting that the lateral ion injection is faster in the face-on dominant DTP-2T than the edge-on dominant DTP-P.

[Comment #15]

Page 14, line 289: [28] is the wrong ref.

[Response #15]

We are grateful for Reviewer #2’s suggested correction. Accordingly, we replaced the reference

in the manuscript as follows.

At page 28:

28. X. Wu et al. High performing solid-state organic electrochemical transistors enabled by glycolated polythiophene and ion-gel electrolyte with a wide operation temperature range from -50 to 110 °C. *Adv. Funct. Mater.* **33**, 2209354 (2023).

[Comment #16]

Page 19, line 415: “In general, the unit cell parameters of typical mixed conducting polymers are the longest along the (100) direction, followed by the (001) and (010) directions (Fig. 6c).” It is very hard to distinguish the polymer chemical structure drawn in the unit cell in fig 6c, making it difficult to tell which orientation direction of the backbone is being discussed (for those who are not crystallography experts).

[Response #16]

We thank Reviewer #2 for his/her constructive comment on polymer chemical structures. Accordingly, we revised the manuscript by adding clear polymer chemical structures in Figure 6 as follows.

Revised Figure 6

End of Responses to Reviewer #2's Comments

Responses to Reviewer #3's Comments

Authors present a study of two closely related organic mixed ionic electronic conductors (OMIEC), denoted DTP-P and DTP-2T. The authors present results from experiments demonstrating that the direction of ionic injection relative to the molecular orientation within the materials affects the performance. Finally, as an important control to rule out other factors, the authors perform tests on samples prepared with different masking schemes to limit the ionic injection direction and demonstrate that the effect persists. The authors present a novel approach to studying OMIECs that sheds light on a previously unexplored aspect of their function and would be of interest to the readers of Nature Communications. However, the authors first need to make some significant revisions to address a lack of clarity in the discussion of the molecular orientation and should then reconsider their interpretation and conclusions. Afterwards, the revised manuscript may be suitable for publication in Nature Communications. Below are detailed comments.

We thank Reviewer #3 for appreciating the value of our research. We did our best to improve the quality of the manuscript in response to his/her comments.

[Comment #1]

The authors prepare thin film transistors with active layers made from the two materials and then apply masking schemes in a new way that limits the direction of ionic injection into the active layers to be either in the plane of the layer (lateral) or out of the plane (vertical). The premise of the article is that the two materials are closely matched in their makeup except for differences in the predominant molecular orientation within the thin films, and that this accounts for the differences in the transient and high-frequency performance of the two materials depending on the ionic current injection direction. Moving front experiments also support this conclusion. All of this places a lot of emphasis on the structural analysis of the materials. The structural evidence comes from X-ray diffraction (XRD) and GIWAXS data presented in Figure 2. The authors need to provide more details of these measurements and to carry out more careful analysis of the data to support their conclusions. Equally importantly,

the authors should use more precise language to state their thesis.

[Response #1]

We are grateful for Reviewer #3's constructive comments. In response to all Reviewer #3's comments, we did our best to improve the quality of our manuscript by providing the details related to measurements and logical reasoning/analysis, and employ precise terminology as follows.

[Comment #2]

P 6-7 discussion of material structure

Lines 115-154

This section needs revision. See also comments below for Figure S2

The authors' analysis of the GIWAXS data is mostly confined to a visual inspection of the 2D data and classification of the two materials' predominant orientation as either "face on" or "edge on". These two descriptions refer to the orientation of the conjugated rings. When the scattering feature from the pi-pi stacking is in the out-of-plane direction (qz), the rings are said to be oriented "face on" (or "plane on"). When the pi-pi stacking feature is oriented along the in-plane direction (qy), the conjugated rings are said to be oriented "edge on". The naming of these two orientation states comes from the conjugated rings, but they also correspond to orientation of the lamellar scattering, arising from the average separation between the polymer backbones as spaced by the side chains. In a face-on system, the lamellar stacking produces scattering features in the in-plane direction while edge-on systems show lamellar scattering in the out-of-plane direction. (DeLongchamp 2011).

In Figure 2b, the orientation of the DTP-2T material does match the classic description of a face-on system. However the first order lamellar peak, expected to have a maximum in the in-plane direction, is mislabeled as (001), corresponding to the repeating unit separation along the backbone, when it should be labeled (100). The very faint feature labeled (300) can be disregarded.

In Figure 2c, the orientation of the DTP-P material is less clear and appears to have a mixture of face-on orientation, with the pi-pi stacking (010) feature strongest in the out-of-plane

direction, and edge-on with the lamellar stacking (n00) strongest in the out-of-plane direction.

In both cases, construction of the pole figure for the (100) and (010) features would give a more quantitative description of the molecular orientation (Savikhin 2018). Integration of the pole figure over different regions of the polar angle χ in the qz vs qr plane could be a way to quantify the fractions of the material that are in the face on vs edge on orientations, which could be useful for normalizing the geometrically normalized time constants further (Figure 5 discussion below).

[Response #2]

We appreciate Reviewer #3's suggested corrections. Accordingly, we corrected (001) to (100) and removed (300) in Figure 2b (DTP-2T). In the case of Figure 2c (DTP-P), however, we suppose that (001) is properly assigned. From the out-of-plane (100) lamellar peak, the lamellar distance was extracted as 17.1 Å, while the in-plane peak was matched to the distance of 15.3 Å. This distance corresponds approximately to the length of quaterthiophene estimated in previous literature (*J. Chem. Phys.* **117**, 321 (2002) 'Spontaneous dissociation of a conjugated molecule on the Si (100) surface') which should be comparable in length to the four aromatic ring repeat unit in our DTP-P system; the repeat unit distance is moreover estimated at 16.3 Å from our energy-minimized structure from DFT. Therefore, we suppose that the in-plane peak of DTP-P could be assigned to (001). Accordingly, we revised the main text by adding pole figures and relevant quantitative analysis in the revised manuscript as follows.

At page 7: It is straightforward to assign the in-plane peak observed from the DTP-2T as (100), while the in-plane peak of DTP-P does not match the lamellar peak distance obtained from the out-of-plane measurements. This in-plane peak is instead attributed to the polymer repeat unit length (001), based on the previous literature³⁵. Therefore, it is not relevant to the dominant molecular orientation of the polymer film. In the 2D GIWAXD pattern of both DTP-P and DTP-2T, a broad ring pattern corresponding to a spacing of ~4.2 Å was found both in-plane and out-of-plane, indicating the presence of SiO₂ substrate³⁶. We also compared the pole figures of the two polymers (Fig. S2) and concluded that DTP-2T predominantly exhibit the face-on orientation whereas DTP-P exhibits mixed orientations, but the edge-on feature is rather dominant over the face-on feature.

At page 28: 36. Kraner, S., Koerner, C., Leo, K. Dielectric function of a poly(benzimidazobenzophenanthroline) ladder polymer, *Phys. Rev. B Condens. Matter* **91**, 195202 (2015).

Figure S2. Pole figures of lamellar (100) and π - π stacking (010) peaks for **a** DTP-2T and **b** DTP-P. Quantitative analysis of the population ratio of edge-on, face-on, and isotropic crystals for **c** DTP-2T (π - π stacking (010) peak) and **d** DTP-P (lamellar-stacking (100) peak).

Figure 2. Crystallographic microstructural analysis of DTP-based mixed ionic–electronic conducting polymers (DTP-P and DTP-2T). **a** XRD patterns of DTP-P and DTP-2T films. The insets illustrate the dominant molecular orientations of DTP-P and DTP-2T, that is, *edge-on*, and *face-on*, respectively, relative to a substrate. 2D GIWAXD pattern images of spin-coated **b** DTP-2T and **c** DTP-P. GIWAXD **d** out-of-plane (q_z) and **e** in-plane (q_{xy}) line-cut profiles of DTP-2T and DTP-P films. **The 1D line-cut profiles were obtained from the 2D GIWAXD regions defined by dashed white boxes.**

[Comment #3]

P12 line 240 “This could be attributed to the edge-on orientation of DTP-P, which is favorable for lateral charge transport”. As discussed above, DTP-P is not purely edge-on but has a mixture of orientations.

[Response #3]

In response to the Reviewer #3’s comment, we revised the main text as follows.

At page 11: **Considering electrochemical doping/dedoping minimally affected the dominant molecular orientations in these polymer films³¹, this trend could be attributed to the edge-on dominant mixed orientations in DTP-P which is still favorable for lateral charge transport compared with the predominant face-on orientation^{37,42}.**

[Comment #4]

P13 line 266 “This clearly indicates that the face-on orientation is highly advantageous for ion drift in lateral direction throughout the mixed conducting polymer film compared to the edge-on orientation.” Note that the lateral direction is parallel to the lamellar stacking direction (100) for the face-on orientation.

[Response #4]

In response to the Reviewer #3’s comment, we revised the main text as follows.

At page 13: This clearly indicates that the face-on orientation is highly advantageous for ion drift in lateral direction (ion drift in parallel with the lamellar stacking direction [100]) throughout the mixed conducting polymer film compared with the edge-on orientation (ion drift in parallel with the π - π stacking direction [010]).

[Comment #5]

P 14 line 300 “This reveals that the ion injection is significantly faster when the ion injection direction is parallel to the ‘face’ of the polymer backbone (i.e., parallel to the lamellar stacking direction). “

The direction “parallel to the lamellar stacking direction” [(100)] is unambiguous. It is in-plane (lateral) for the predominantly face on DTP-2T and predominantly out-of-plane (vertical) for the mixed DTP-P. The direction “parallel to the ‘face’ of the polymer backbone” is unclear. The direction of any Bragg reflection is perpendicular to the lattice planes that cause the reflection. If these lattice planes are what is being referred to as the ‘face’ of the polymer backbone, then more explanation is needed to make this clear. Note that these planes contain the polymer backbone and also the sidechains, and that the lamellar stacking direction is perpendicular to the length of the polymer backbone. [Verploegen 2010]

[Response #5]

We are grateful for Reviewer #3’s constructive comments on the description of ion injection direction. Accordingly, we revised the main text as follows.

At page 14: This reveals that the ion injection is significantly faster when the ion injection direction is in parallel with the lamellar stacking direction [100].

[Comment #6]

P 15 line 315 “Although DTP-2T and DTP-P showed that ionic mobilities and frequency-dependent transient behaviors of OECTs depend on the direction of ion injection, there exists still the possibility of material-dependent characteristics (e.g., molecular weight, relative crystallinity) regardless of the molecular orientation of OMIECs. Finally, different patterning methods were employed to demonstrate the molecular orientation dependency of the transient

behavior.” This is a great control experiment to finesse the details of the molecular orientation....

[Response #6]

We thank Reviewer #3 for appreciating the given control experiments in our research.

[Comment #7]

P 16 Line 347 “The DTP-2T and DTP-P devices fabricated by parylene patterning showed the geometry-normalized τ values of 9.9×10^8 and 5.4×10^8 s/m², respectively, which suggests the transient response of DTP-P device is faster. The DTP-2T and DTP-P devices fabricated by CYTOP patterning showed the values of 5.7×10^8 and 1.0×10^9 s/m², respectively, revealing that this trend is opposite to those patterned with parylene.” However, the interpretation still relies on comparisons of performance of the 2 different materials. Isn’t it more interesting to make performance comparisons of the same material with the ion injection constrained in 2 different directions? Should these 4 values (with appropriate error bars) for the geometrically-normalized time constants be highlighted in a figure? Something similar to 5c or 5e, but grouped differently. There would be 2 adjacent bars for $\tau_{\text{norm}}(\text{DTP-2T, parylene})$ and $\tau_{\text{norm}}(\text{DTP-2T, CYTOP})$ and also 2 adjacent bars for $\tau_{\text{norm}}(\text{DTP-P, parylene})$ and $\tau_{\text{norm}}(\text{DTP-P, CYTOP})$. Above the first pair could be an illustration showing the lamellar stacking direction for (face-on) DTP-2T (in-plane or horizontal) and above each bar could be an indication of the ionic injection direction. And similar for the second pair, but for edge-on DTP-P with the vertical lamellar stacking.

[Response #7]

We thank Reviewer #3 for his/her constructive suggestion to improve the quality of this study. As Reviewer #3 pointed out, we calculated geometry-normalized time constants of DTP-P and DTP-2T devices fabricated by parylene and CYTOP patterning and added the relevant discussion in the original manuscript (but no their plots/representation in the main figure set, though) for fair comparison: DTP-P parylene (5.4×10^8 s m⁻²), DTP-P CYTOP (1.0×10^9 s m⁻²), DTP-2T parylene (9.9×10^8 s m⁻²), and DTP-2T CYTOP (5.7×10^8 s m⁻²). Accordingly, we revised Figure 5 as follows with the reviewer’s suggestion now included in Figure 5f.

Figure 5. Molecular-orientation-dependent transient behavior of DTP-2T and DTP-P. **a** schematic representation of OEET devices fabricated by different patterning methods. Black arrows indicate ion injection direction when the gate bias is applied to the OEET devices. The active material of the two devices on the top panel (blue) represents a face-on dominant molecular orientation, and that of the two devices on the bottom panel (red) represents an edge-on dominant molecular orientation. **b** Transient response of the drain current at a constant V_D of -0.6 V and a square voltage pulse of 10 s ($V_G = -0.8$ V) applied at the gate electrode. The devices were fabricated by the CYTOP patterning method. **c** Plot of rise time constant obtained from transient response of DTP-2T- and DTP-P-based OEETs fabricated by parylene and CYTOP patterning methods. The measurements were performed using more than five devices for each set. **d** Plot of frequency-dependent normalized transconductance obtained from the frequency response measurements of the DTP-2T- and DTP-P-based OEETs fabricated by parylene and CYTOP patterning methods. Biasing condition was $V_D = -0.6$ V, and sine wave V_G with varying frequency was used. **e** Transconductance obtained from the DTP-2T- and DTP-P-based OEETs by the application of V_G sine wave with a frequency of 10 Hz. **f** Geometry-normalized τ values obtained from the DTP-2T and DTP-P-based OEETs fabricated by parylene and CYTOP patterning methods.

At page 17: The DTP-2T and DTP-P devices fabricated by parylene patterning showed the geometry-normalized τ values of $9.9 \pm 1.7 \times 10^8$ and $5.4 \pm 0.3 \times 10^8$ s/m², respectively, which suggests the transient response of DTP-P device is faster (Fig. 5f). The DTP-2T and DTP-P devices fabricated by CYTOP patterning showed the values of $5.7 \pm 0.2 \times 10^8$ and $10 \pm 1.6 \times 10^8$ s/m², respectively (Fig. 5f), revealing that this trend is opposite to those patterned with

parylene. Regarding ion injection constrained in two distinct directions, the DTP-P-based OECT exhibits a faster geometry-normalized transient response upon parylene patterning than upon CYTOP patterning whereas the opposite is true for the DTP-2T-based OECT.

[Comment #8]

Another question: No error bars are given for the geometrically normalized time constants, but it looks like the two fastest time constants and the two slowest time constants are almost the same. Is this result significant? Or just a coincidence? I'm asking because besides the geometry, the other normalization that is needed is to normalize with respect to the molecular orientation. DTP-2T seems strongly face on while DTP-P is more mixed. If the normalization also took into account the mean orientation, would the time constants agree quantitatively?

[Response #8]

In response to Reviewer #3's comment on the statistical significance of different time constants, we discussed this matter in the revised main text. Regarding the normalization over molecular orientation, we should admit that it is beyond our expertise since it requires more systematic material characterizations in terms of molecular orientation, paracrystallinity, etc. and more judicious experimental design. Accordingly, we revised the main text as follows to discuss this matter and mention possible limitations on this research theme.

At page 16: The DTP-2T and DTP-P devices fabricated by parylene patterning showed the geometry-normalized τ values of $9.9 \pm 1.7 \times 10^8$ and $5.4 \pm 0.3 \times 10^8$ s/m², respectively, which suggests the transient response of DTP-P device is faster. The DTP-2T and DTP-P devices fabricated by CYTOP patterning showed the geometry-normalized τ values of $5.7 \pm 0.2 \times 10^8$ and $10 \pm 1.6 \times 10^8$ s/m², respectively (Fig. 5f), revealing that this trend is opposite to those patterned with parylene.

At page 17: We note that these geometry-normalized time constant values in this study suggest that the molecular orientation is one of the important factors influencing time constants. It is, however, not the only factor, considering that paracrystallinity, crystallite size, and molecular weight may affect the apparent capacitance and resistance of the ionic circuit between the channel and the gate electrode and, thereby, the extracted time constant values⁴³. Decoupling all these effects on the time constant value would be a future research theme worth exploring.

[Comment #9]

P 16 line 351 ” Therefore, it can be deduced that the transient response is relatively fast when the ion injection direction is parallel to the polymer backbone orientation, while the transient response is relatively slow when the ion injection direction is not parallel to the polymer backbone orientation.” Authors should use more precise language here. The direction along the polymer backbone should be the (001) direction.

[Response #9]

In response to the Reviewer #3’s comment, we revised the manuscript as follows.

At page 17: Therefore, it can be deduced that the transient response is relatively fast when ions are *vertically* injected into edge-on-dominant films compared with that into face-on-dominant films (Fig. 5a lower left column), while the transient response is relatively slow when ions are *laterally* injected into edge-on-dominant films compared with that into face-on-dominant films (Fig. 5a lower right column).

[Comment #10]

I believe the data presented in Figure 5, together with correct interpretation of the structural data in Figure 2 means that the opposite is true. The transient response is relatively fast when the ion injection direction is parallel to the lamellar stacking direction, which means when the ion injection is perpendicular to the polymer backbone. Equivalently, the transient response is relatively fast when the ion injection direction is perpendicular to the pi-pi stacking direction. P 18 line 392 “ii) the ions can drift through the lamellar gap (the gap between polymer backbones arranged along the (100) direction) due to the presence of hydrophilic glycol side chains and the large distance of the lamellar gap;” As discussed above, the evidence suggests ionic injection is faster when it is parallel to the lamellar stacking direction. The lamellar stacking direction is perpendicular to the lamellae. Moving in the (100) direction means moving along the glycol side chains.

[Response #10]

We are deeply grateful for Reviewer #3's appreciation of our research and constructive suggestion. For the clear understanding of the claims in this research in terms of terminology and expressions related to directionality, we revised the main text as follows.

At page 14: This reveals that the ion injection is significantly faster when the ion injection direction is in parallel with the lamellar stacking direction [100].

At page 17: Therefore, it can be deduced that the transient response is relatively fast when ions are vertically injected into edge-on-dominant films compared with that into face-on-dominant films (Fig. 5a left column), while the transient response is relatively slow when ions are laterally injected into edge-on-dominant films compared with that into face-on-dominant films (Fig. 5a right column).

At page 18: As demonstrated from the transient measurements, the doping efficiency is higher when the ion injection direction is parallel with the lamellar stacking direction [100] (i.e., the higher the doping efficiency, the higher is the g_m value).

[Comment #11]

P18 line 389 “i) ions may infiltrate into the gap of hydrophobic π - π stacks but their drift velocity is very low due to the relatively large solvation shells of hydrated ions, thus the effect of the overall ion drift through the crystalline polymer domain could be ignored^{19,25,43}. due to the relatively large hydrated ions,” In contrast to the ‘lamellar gaps’ mentioned above, the ions do move more quickly when they move perpendicular to the pi-pi stacking direction, so they do move more rapidly when they encounter the ‘pi-pi gaps’.

[Response #11]

In this part, the term ‘infiltration’ refers to ions literally passing through the gaps. Therefore, in the sentence mentioned by Reviewer #3, we are asserting that the reason why ‘the ions do move more quickly when they move perpendicular to the pi-pi stacking direction’ is not because they are moving through the pi-pi gaps, but rather due to the shorter ion movement pathway when the ions encounter the pi-pi gaps, considering unit cell of the crystallized domain of polymer (Figure 6c). The intention of the statement at page 18 line 389 was to highlight that we disregard

the ion drift through the pi-pi gap (i.e., 'infiltration') because hydrated ions are much larger than the pi-pi gap distance. Accordingly, we have revised the main text for clarity as follows.

At page 19: From the abovementioned results, we were able to discuss how ions drift through the OMIEC films with different molecular orientations. First, we assumed three basic principles for the following discussion: i) There exist two distinct ion drift behaviors when ions encounter crystalline domains, namely, *infiltration through* and *bypassing around* crystalline domains but, in this research, the overall ion drift through the π - π gap (i.e. in-between two pi-stacked polymer backbones) could be ignored due to the relatively large-size hydrated ions (hydrated Cl^- ions $\sim 7.24 \text{ \AA}$, π - π distance $\sim 3.7 \text{ \AA}$)^{19,25,44}.

[Comment #12]

P 19 Line 406

“the drift velocity of ions through crystalline domain is very low because the size of the hydrated ion is larger than π - π stacking distance” It would be helpful for the authors to make the comparison of these sizes explicitly in the text.

[Response #12]

In response to Reviewer #3's comment, we revised the main text as follows.

At page 19: i) There exist two distinct ion drift behaviors when ions encounter crystallized domains, namely, *infiltration through* and *bypassing around* crystalline domains but, in this research, the overall ion drift through the crystalline polymer domain could be ignored **due to the relatively large-size hydrated ions (hydrated Cl^- ions $\sim 7.24 \text{ \AA}$, π - π distance $\sim 3.7 \text{ \AA}$)**^{19,25,44}.

[Comment #13]

P 19 418-420 “Assuming that there exists densely packed polymer crystal consisting of many unit cells, and that the number of unit cell along with each axis ([100], [010]) is identical regardless of axis, the coherence lengths of (100) and (010) planes will be proportional to the length of each unit cell parameter” Instead of making this assumption, authors should analyze the width of the (010) and (100) peaks and obtain the mean number of unit cells contributing

to the diffraction, according to the Debye-Scherrer theory, taking into account the experimental broadening from the use of area detectors [Smilgies 2009]

[Response #13]

In response to Reviewer #3's comment, we analyzed the full width at half-maximum value to estimate the number of unit cells contributing to the diffraction. Accordingly, we revised our manuscript as follows.

At page 20: To deal with the different effects of the dominant molecular orientation, we must consider the general structure of crystallized polymeric materials. As shown in Supporting Information (Table S4), the coherence lengths of the (100) and (001) planes are significantly longer than that of the (010) plane. Although the unit chain length of DTP-2T is undiscernible due to the absence of a (001) peak, it could be inferred that the coherence length of the (001) plane would be similar to or longer than that of (100) plane by considering the molecular structure and previously reported literatures^{13,14,15,20,37,46,47,48,49,50,51}. Therefore, when the ion injection direction is in parallel with the π - π stacking direction [010], the *bypassing* pathway of ions becomes significantly longer than when the ion injection direction is in parallel with the lamellar stacking [100] or unit chain length [001] direction (Fig. S14 and Fig. 6d to g, see Figure 6c for clear schematic representation of ion injection direction).

At page 35 of Supporting Information:

Table S4. Solid-state packing parameters extracted from the GIWAXD measurements.

Polymer	Lamellar stacking (100)			π - π stacking (010)			Unit chain length (001)		
	q [\AA^{-1}]	d [\AA]	L_c [\AA]	q [\AA^{-1}]	d [\AA]	L_c [\AA]	q [\AA^{-1}]	d [\AA]	L_c [\AA]
DTP-2T	0.323	19.5	76.1	1.711	3.7	33.8			
DTP-P	0.373	17.1	62.6	1.439	4.4	10.9	0.412	15.3	86.7

[Comment #14]

P 23 GIWAXD methods section should describe the experiment more fully.

Authors should provide a reference describing the beamline and instrument used in this study.

Line 490 "c5" should be the Angstrom symbol.

Authors should provide the dimensions of the samples, the size of the incident beam, the divergence of the beam. These parameters contribute toward evaluating the experimental resolution. Authors should also report typical exposure times, and the environment of the sample during the measurement (vacuum? He? Ambient?). The authors should also define the regions of integration used for the linecuts presented in Figures 2d, e.

[Response #14]

In response to Reviewer #3's comment, we revised the experimental section as follows.

At page 24: GIWAXD measurements were conducted at the 9A U-SAXS beamline of the Pohang Light Source (PLS), Republic of Korea. All samples for GIWAXD were prepared by spin-casting of polymer solution (5 mg/mL) on p-Si⁺⁺/SiO₂ (300 nm) substrates. The substrate size was 15 mm × 15 mm while the thickness of polymer film was set at ~50 nm. The wavelength of X-rays was 1.12370 Å ($E = 11.025$ keV), and the incidence angle of the beam light was ~0.1°. The size of the incident beam was 60 μm × 298 μm (vertical and horizontal) at detector position, and the divergence of the beam was 25 / 45 urad (vertical and horizontal). Typical exposure time was ~12 s, and all measurements were conducted in ambient conditions. The images from GIWAXD were obtained with a 2D FT-CCD module of Rayonix MX170-HS, while the sample-to-detector distance was adjusted to be 225 mm.

Figure 2. Crystallographic microstructural analysis of DTP-based mixed ionic–electronic conducting polymers (DTP-P and DTP-2T). a XRD patterns of DTP-P and DTP-2T films. The insets illustrate the dominant molecular orientations of DTP-P and DTP-2T, that is, *edge-*

on, and *face-on*, respectively, relative to a substrate. 2D GIWAXD pattern images of spin-coated **b** DTP-2T and **c** DTP-P. GIWAXD **d** out-of-plane (q_z) and **e** in-plane (q_{xy}) line-cut profiles of DTP-2T and DTP-P films. **The 1D line-cut profiles were obtained from the 2D GIWAXD regions defined by dashed white boxes.**

[Comment #15]

P 30 Figure 2

Figure 2a. (1) The pseudocolor scale bars in the GIWAXD data show a logarithmic scale that makes the lamellar diffraction peaks ($n00$) for the DTP-P material (2c) plainly visible up to 3rd order. However, the XRD data appear to be on a linear scale and only the first order lamellar peak is visible. If the data were on a log scale, would the higher order peaks also be visible? Or is the data too noisy to show the higher order peaks? Perhaps longer integration times would reveal these features. No details about the x-ray exposure times or background subtraction are given in the experimental description on p. 23. (2) The intensity is plotted as a function of the scattering angle, 2θ . It would be more meaningful to convert this to the equivalent $q_z = (4\pi/\lambda)\sin(\theta)$, which would allow more direct comparison with the GIWAXD data. (3) Figure 2, c. 2D data should be reshaped in the representation q_z vs $q_r = \sqrt{q_x^2 + q_y^2}$, which would make it clear that a wedge shaped region is not observed in the GIWAXD measurement [Rivnay 2012].

[Response #15]

(1) As shown in the Figure R5, the higher order peaks were not visible even though the data are on a log scale. Considering that the quality of XRD spectra could not be improved even after increasing exposure time and conducting background subtraction, we suppose that these rather noisy results are due to the limited sensitivity/resolution in our XRD measurement setup. Accordingly, we revised the main text as follows.

At page 24: For XRD spectra, the polymer films prepared on Si substrates were employed. The XRD spectra were collected using an X-ray diffractometer (RigakuD/max-2500) with $\text{Cu K}\alpha$ radiation ($\lambda = 1.54 \text{ \AA}$) at 40 kV and 100 mA. **Exposure time was set as 1 s and the background peaks were subtracted.**

Figure R5. XRD patterns of DTP-2T and DTP-P film plotted with log scale.

(2) According to the Reviewer #3's comments, we revised Figure 2 as follows. In Figure 2a, x axis is converted to q_z , which allows for a more straightforward comparison with GIWAXD patterns.

(3) The 2D GIWAXD patterns are reshaped to remove the wedge-shaped region.

Figure 2. Crystallographic microstructural analysis of DTP-based mixed ionic–electronic conducting polymers (DTP-P and DTP-2T). **a** XRD patterns of DTP-P and DTP-2T films. The insets illustrate the dominant molecular orientations of DTP-P and DTP-2T, that is, *edge-on*, and *face-on*, respectively, relative to a substrate. 2D GIWAXD pattern images of spin-coated **b** DTP-2T and **c** DTP-P. GIWAXD **d** out-of-plane (q_z) and **e** in-plane (q_{xy}) line-cut profiles of DTP-2T and DTP-P films. **The 1D line-cut profiles were obtained from the 2D GIWAXD regions defined by dashed white boxes.**

[Comment #16]

P 34 Figure 5

A. Presents schematics of the different mask patterning approaches applied to idealized face on and edge on materials. (1) Judging from the schematic of the OMIEC unit cell in Figure 6C, which shows the (100) direction as the long direction, the edge on material is not illustrated correctly. For edge on material, the (100) direction is vertical, so the red tiles in the bottom panels of columns 2 and 4 should have their long dimensions oriented vertically (2) The labels on the 2nd and 4th columns are inconsistent. “Vertical” refers to the lab/substrate frame of reference. For the Parylene patterning (second column), the ion injection is always vertical. For the CYTOP patterning (4th column), the ion injection is always “lateral” (or horizontal). So the label in the fourth column, second row is incorrect. On the other hand, “parallel” refers to a comparison of two vectors, one of which is the ion injection direction. The other is not clear. From the illustration, in the 4th column top panel, “parallel injection” means injection parallel to the lamellar stacking (100) direction, which is correct. Meanwhile, as illustrated in the 2nd column bottom panel, “parallel injection” means parallel to the backbone repeat direction (001) which is incorrect. The red tiles should be oriented so their long dimension is vertical, then the (100) direction will be (anti)parallel to the injection.

[Response #16]

We appreciate Reviewer #3’s concern. Accordingly, we revised Figure 5 to clearly describe the suggested molecular orientations and ion injection directions.

Figure 5. Molecular-orientation-dependent transient behavior of DTP-2T and DTP-P. **a** schematic representation of OEFT devices fabricated by different patterning methods. Black arrows indicate ion injection direction when the gate bias is applied to the OEFT devices. The active material of the two devices on the top panel (blue) represents a face-on dominant molecular orientation, and that of the two devices on the bottom panel (red) represents an edge-on dominant molecular orientation. **b** Transient response of the drain current at a constant V_D of -0.6 V and a square voltage pulse of 10 s ($V_G = -0.8$ V) applied at the gate electrode. The devices were fabricated by the CYTOP patterning method. **c** Plot of rise time constant obtained from transient response of DTP-2T- and DTP-P-based OEFTs fabricated by parylene and CYTOP patterning methods. The measurements were performed using more than five devices for each set. **d** Plot of frequency-dependent normalized transconductance obtained from the frequency response measurements of the DTP-2T- and DTP-P-based OEFTs fabricated by parylene and CYTOP patterning methods. Biasing condition was $V_D = -0.6$ V, and sine wave V_G with varying frequency was used. **e** Transconductance obtained from the DTP-2T- and DTP-P-based OEFTs by the application of V_G sine wave with a frequency of 10 Hz. **f** Geometry-normalized τ values obtained from the DTP-2T and DTP-P-based OEFTs fabricated by parylene and CYTOP patterning methods.

[Comment #17]

Supporting Material, p 14, Figure S2

The 2D data shows a maximum in the feature at $q \sim 1.5 \text{ \AA}^{-1}$ not in the in-plane or out-of-plane directions, but at 37.5 deg measured from the in-plane direction. The diagram below the data is intended to aid in interpreting the origin of this feature. However the interpretation presented (see main text, p 7) confuses concepts to arrive at an erroneous interpretation. Consideration of the angle of the maximum in the scattering data to obtain a molecular tilt angle as in Figure S2 comes from 2D crystallography of amphiphilic monolayers, as described by [Kaganer 1999], relies on the intersection of the reciprocal disk representing the form factor of rod-like molecules and the six Bragg rods in the reciprocal lattice representing a hexagonally packed 2D crystal when the rods are tilted (see Figure 5 in [Kaganer 99] and discussion in text). Following the authors' logic, this feature would not correspond to the lamellar spacing, but would correspond to the mean spacing between the ethylene-glycol chains. However, this cannot be the case because only 1 such feature is observed. As stated in [Kaganer 99], the tilting of the rods breaks the symmetry, and so additional peaks due to the crystalline EG chains should also be observed at different q values. Instead, this feature appears to be due the broad distribution of the pi-pi stacking in the material.

[Response #17]

We are grateful for Reviewer #3's suggested correction and have incorporated them into our revised manuscript. As recommended by Reviewer #3, we have made the necessary changes, including the removal of the description regarding the tilted angle. However, during the course of the revision process, we have come to the conclusion that observed broad ring pattern in both DTP-2T and DTP-P can be attributed to the presence of SiO₂ substrate in our samples. While we acknowledge Reviewer #3's suggestion that this pattern is linked to the broad distribution of the pi-pi stacking, we find it important to note that this interpretation contradicts the pi-pi stacking peak observed in the out-of-plane DTP-2T and the lamellar stacking peak observed in the out-of-plane DTP-P. Therefore, our revised manuscript now reflects the following changes.

At page 7: Therefore, it is not relevant to the dominant molecular orientation of the polymer film. In the 2D GIWAXD pattern of both DTP-2T and DTP-P, a broad peak corresponding to a spacing of $\sim 4.2 \text{ \AA}$ was found, indicating the presence of SiO₂ substrate³⁶. We also compared

the pole figures of two polymers (Fig. S2) and concluded that DTP-P exhibit mixed orientations but the edge-on feature is rather dominant over the face-on feature.

At page 28: 36. Kraner, S., Koerner, C., Leo, K. Dielectric function of a poly(benzimidazobenzophenanthroline) ladder polymer, *Phys. Rev. B Condens. Matter* **91**, 195202 (2015).

=====
End of Responses to Reviewer #3's Comments
=====

REVIEWERS' COMMENTS

Reviewer #2 (Remarks to the Author):

The authors responded to my comments sufficiently and this version is now very interesting to read.

Reviewer #4 (Remarks to the Author):

I think the authors have addressed the reviewers's questions and comments,thus the current version is recommended to be publihsed in Nature Commu.

=====
Responses to Reviewer #2's comments
=====

The authors responded to my comments sufficiently and this version is now very interesting to read.

We deeply appreciate Reviewer #2's positive reception of our research. Thank you for the constructive comments and suggestions that have greatly improved our manuscript.

=====
End of Responses to Reviewer #2's Comments
=====

Responses to Reviewer #4's Comments

I think the authors have addressed the reviewers' questions and comments. Thus, the current version is recommended to be published in Nature Communications.

We are sincerely thankful for Reviewer #4's recognition of our efforts. Your kind consideration of our response to Reviewers is truly appreciated.

End of Responses to Reviewer #4's Comments